# Sustainable Strategies for the Control of Pests in Coffee Crops

Carmenza E. Góngora *, Zulma Nancy Gil, Luis Miguel Constantino and Pablo Benavides

Department of Entomology, National Coffee Research Center, Cenicafé, Manizales 170009, Colombia; zulma.gil@cafedecolombia.com (Z.N.G.); luismiguel.constantino@cafedecolombia.com (L.M.C.); pablo.benavides@cafedecolombia.com (P.B.)
* Correspondence: carmenza.gongora@cafedecolombia.com; Tel.: +57-3154869466

**Abstract:** Coffee is a worldwide commodity, and both coffee-producing and coffee-consuming countries have real concerns about environmental problems and economic growth strategies based on the efficient use of resources. Because this crop is a perennial, pests can significantly affect coffee production, causing considerable yield losses and threatening coffee supply and security. The presence of insects and control strategies for coffee pests is becoming a challenge. Environmental sustainability, conservation of biodiversity, and safety of the coffee seed must go hand in hand with the economic sustainability of coffee growers. This is especially important, as there has been an increase in demand for coffee and new consumer interest in differentiated quality coffee. Regular pest control methods based only on the use of synthetic pesticides are no longer effective or sustainable due to the development of insecticide resistance and negative effects on the environment, human health, and biodiversity. Thus, to ensure better control and ecological sustainability, it is crucial to reduce pesticide use by adopting original alternative strategies to maintain pest populations below the economic threshold level and towards reaching the European Green Deal. In this review, we collect information available for sustainable control of the principal coffee pests in Colombia: coffee berry borer (CBB), *Hypothenemus hampei*; *Monalonion velezangeli*; coffee root mealybugs; coffee leaf miner: *Leucoptera coffeella*; and the coffee red spider mite: *Oligonychus yothersi.* The control strategies include deep knowledge of the biology of insects and the coffee plant, their relationship with weather and habitats, as well as natural controllers. These control strategies do not involve the use of insecticides, are ecologically friendly and novel, and can be applied in other coffee-producing countries.

**Keywords:** coffee berry borer; CBB; *Hypothenemus hampei*; *Monalonion velezangeli*; coffee root mealybugs; coffee leaf miner; *Leucoptera coffeella*; coffee red spider mite; *Oligonychus yothers*

## 1. Introduction

Coffee cultivation in Colombia has a history of around 300 years since it was brought to the country in the 19th century. In 1835, the first coffee bags produced (60 K of seed) in the eastern zone of the country were exported, and by 1850 coffee plantations had reached the departments of Cundinamarca, Antioquia, and Caldas. By the end of the 19th century, production had increased from 60,000 bags to more than 600,000 and coffee had become the main agricultural export product in the country, a position it still holds today [1].

Unlike other producing countries, in Colombia, coffee growers are smallholders who live on their farms and grow other food products alongside coffee. This characteristic means that insecticides have generally not been used indiscriminately due to the presence of people. Coffee cultivars have been kept free of important insect pests since the beginning of their development as a commercial exploitation. It is recognized that Colombia is the only country in the world where coffee plantations are managed without or with very low use of insecticides [2].

Additionally, being a non-native crop that originated from Africa, its pests arrived late in Colombia. The main one is the coffee berry borer (CBB), which invaded in 1988 [3]. Other arthropods have not become serious pests due to the fact that these agroecosystems

are quite stable with great biodiversity, which favors the development of beneficial agents and maintains potential pests present on farms in equilibrium.

In general, in Colombia, the coffee crop has been considered clean and environmentally friendly due to the low level of pesticide use in the 842,000 coffee hectares planted in the country (Official presentation of National Federation of Coffee Growers, 2022). Firstly, coffee leaf rust has been managed with the use of resistant varieties in 86% of the coffee hectares (724,120 hectares corresponds to these varieties). Secondly, control of the coffee borer has been based on a nationwide integrated pest management (IPM) program, in which knowledge of climate conditions, together with cultural practices and crop agronomic management, have maintained this pest below economic threshold levels.

The other pests that can be found in Colombia are *Monalonion velezangeli*; coffee root mealybugs; coffee leaf miners, *Leucoptera coffeella*; and the coffee red spider mite, *Oligonychus yothersi* (McGregor) (Acari: Tetranychidae). For all these insect pests, we are proposing new control strategies that do not involve the use of insecticides and are ecologically friendly and novel. Depending on the physiology of the crop and weather conditions, they can be applied in other coffee-producing countries.

## 2. Coffee Berry Borer (CBB) in Colombia

The coffee berry borer (CBB), *Hypothenemus hampei* (Ferrari, 1867) (Coleoptera: Curculionidae: Scolytinae) invaded Colombia during the years of 1988 and 1993, reaching infestation levels of coffee beans over 25% nationwide [3]. This insect pest is considered the only key pest of coffee crops in Colombia.

### 2.1. Insect Biology

Inside of the coffee seed, the CBB adult female emerges from its pupa and mates with its sibling male. The sex ratio is 10:1 (female/male), and the females start laying eggs after three days. The oviposition period lasts 20 days, and females lay between two and three eggs per day. At a temperature of 21 °C, the egg incubation lasts 9 days, the larva stage 19 days, the pupa stage 11 days, and the myelinization of the adult 7 days. The total cycle from egg to adult is estimated to be 45 days, whereas at a temperature of 18 °C, the cycle can last 60 days (Figure 1) [4].

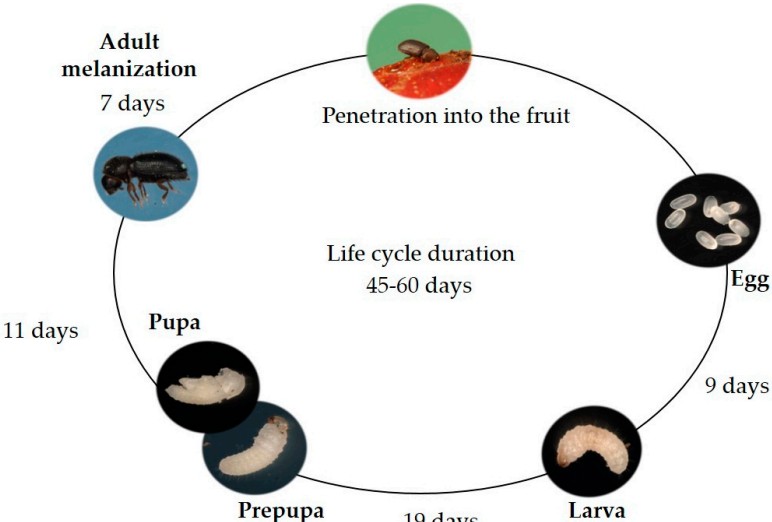

**Figure 1.** *Hypothenemus hampei* life cycle in coffee at 21 °C.

### 2.2. Insect Damage

The damage caused by the insect occurs once the female infests the coffee berry (Figure 2). An IPM strategy has been developed, validated, constantly updated, and taken in place since 1996, marking the breakdown of the field population and keeping damages

under threshold levels of 5% (Figure 3). Unfortunately, climate change, especially El Niño ENOS events, triggers CBB population dynamics (Figure 4); as a consequence, this increases infestation levels over action thresholds of 2%, which pushes chemical control, increasing control costs and affecting coffee quality and productivity.

CBB is a coffee insect pest affected mainly by temperature. The lower the altitude, the higher the temperature and the larger the CBB population. The number of CBB generations related to temperature (Figure 5) (Adapted from Cenicafe, Centro Nacional de Investigaciones del café, Informe Anual de Labores 2020, pg. 63 https://doi.org/10.3 8141/10783/2020) has allowed the construction of CBB vulnerability maps to the whole coffee-growing area of Colombia under the three ENSO conditions: La Niña, neutral, and El Niño (Figure 6) (Adapted from Cenicafe, Informe Anual de Labores 2021, pg. 49 https://doi.org/10.38141/10783/2021). This information points to the worst scenario for CBB in Colombia during El Niño, in which 11% of the coffee-growing area enters in a high vulnerability state, while 49% becomes moderate; only 40% of the entire coffee area may be considered to be under no CBB threat during this period (Table 1).

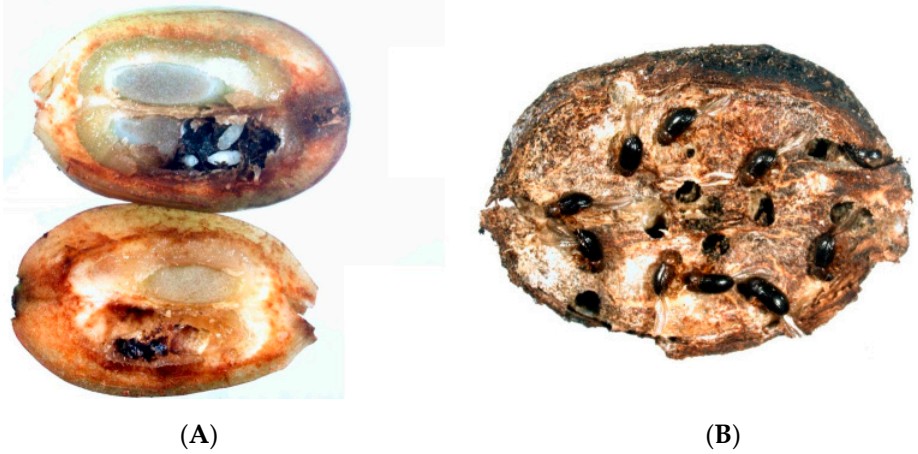

(**A**)　　　　　　　　　　　　　　　　(**B**)

**Figure 2.** *Hypothenemus hampei* damage (**A**) Larvae consuming both seeds; (**B**) Totally damaged seed and adults.

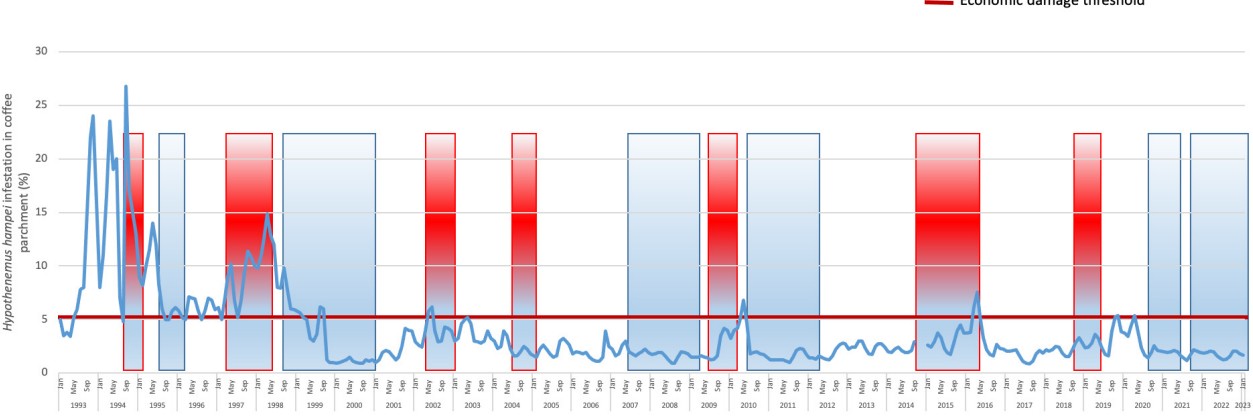

**Figure 3.** CBB Infestation levels in exporting coffee parchment (source: FNC, Almacafé, the Colombian Coffee Storage company). The blue line is the *H. hampei* infestation, it is indicated in the X axes.

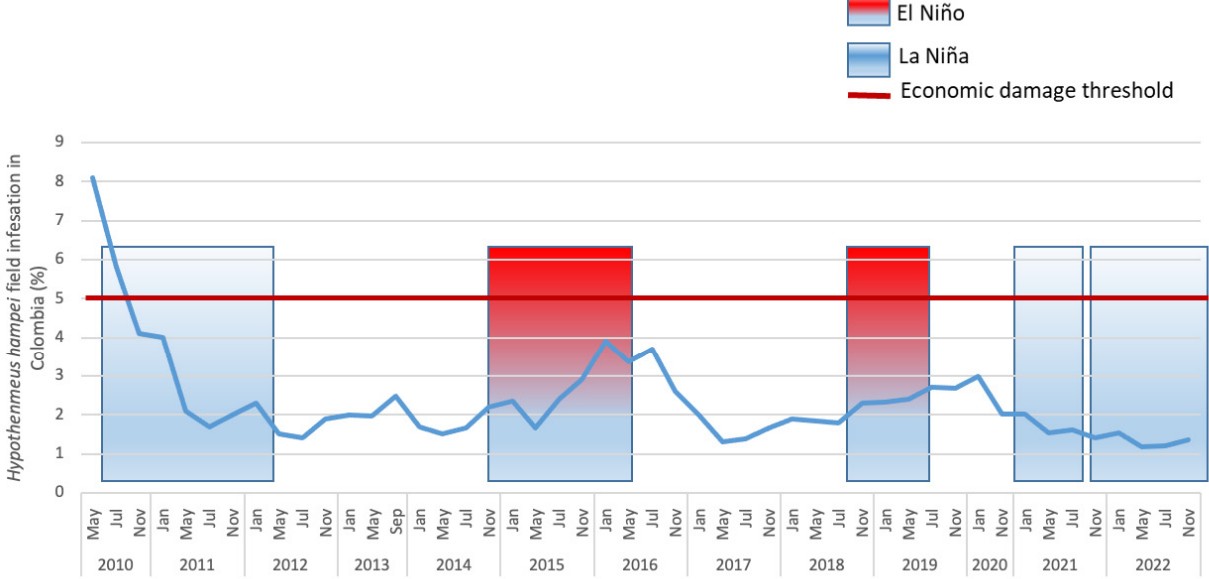

**Figure 4.** CBB field infestation levels in Colombia (source: Technical Management Office, Colombia Coffee Growers Federation). The blue line is the *H. hampei* infestation, it is indicated in the X axes.

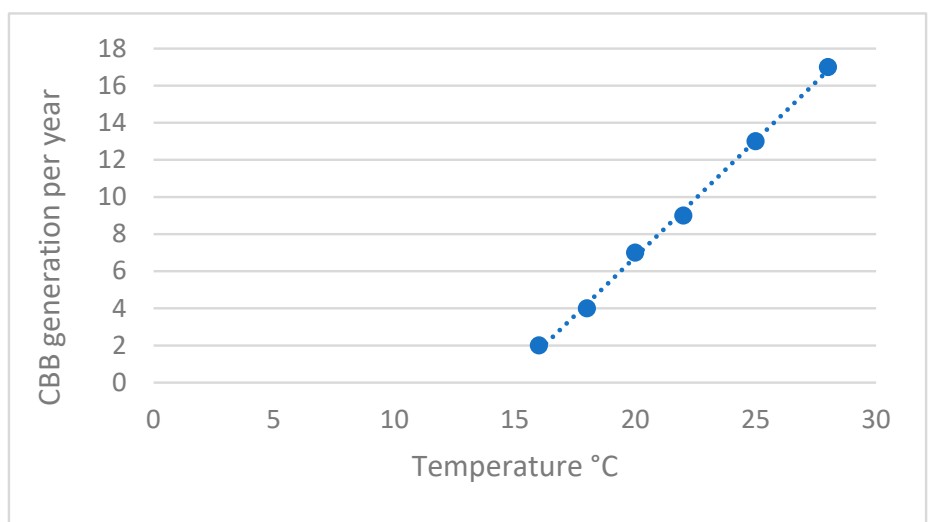

**Figure 5.** Potential number of CBB generations as a function of temperature according to equation NG = −18.1273 + 1.2462 T, where NG: number of generations and T: average annual temperature.

**Table 1.** Percentage of the coffee-growing area of Colombia vulnerable to the coffee berry borer depending on ENSO events.

| ENSO | Coffee Growing Area (%) | | | |
|---|---|---|---|---|
| EVENT | Very Low | Low | Moderate | High |
| Neutral | 11 | 40 | 43 | 6 |
| La Niña | 20 | 45 | 32 | 3 |
| El Niño | 6 | 34 | 49 | 11 |

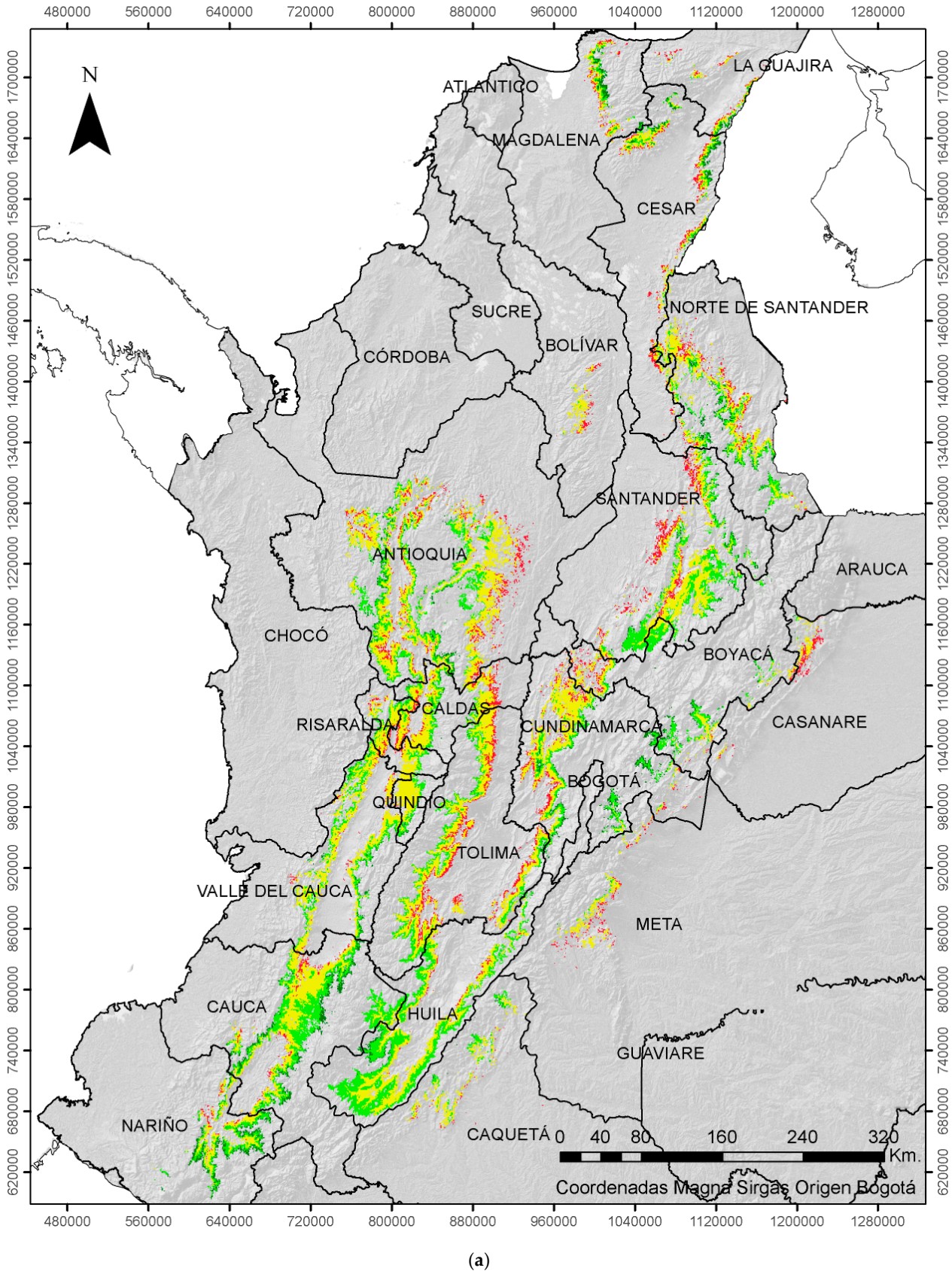

(**a**)

**Figure 6.** *Cont.*

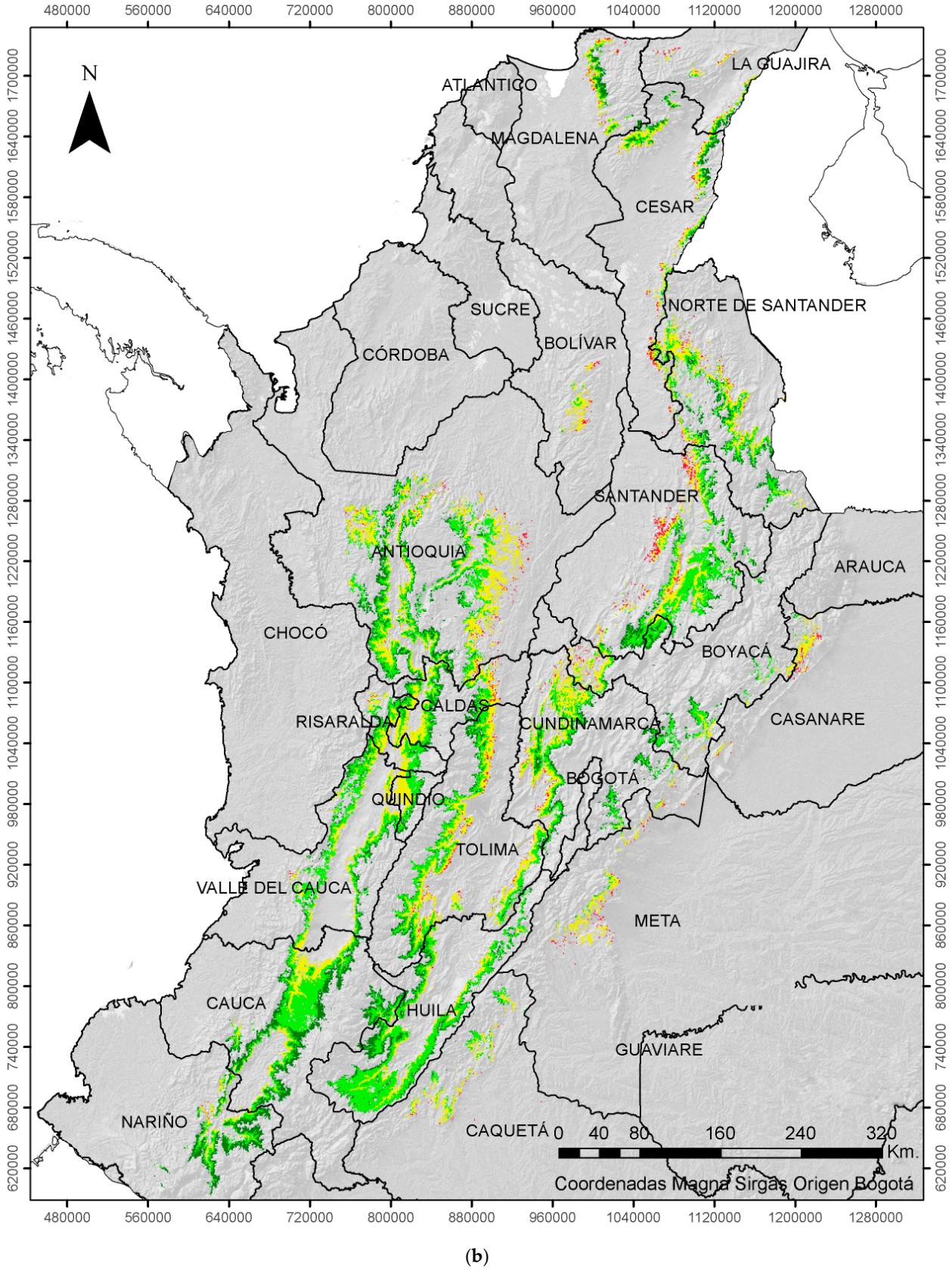

(**b**)

**Figure 6.** *Cont.*

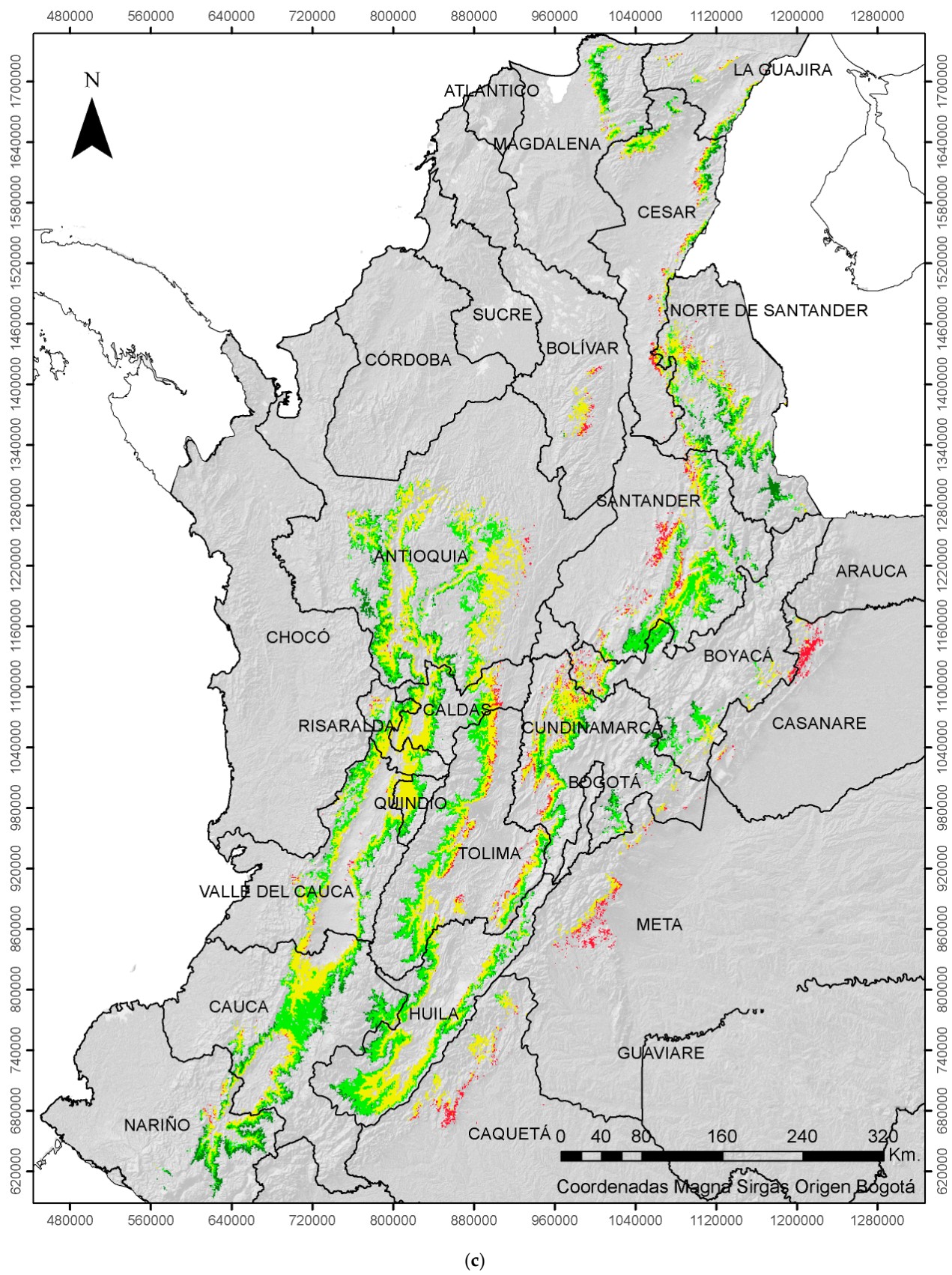

**Figure 6.** Vulnerability map of the Colombian coffee zone to the coffee berry borer in a year with El Niño (**a**), La Niña (**b**), and Neutral (**c**) ENSO scenarios. Dark green corresponds to very low vulnerability and light green, yellow, and red to low, moderate, and high vulnerability, respectively.

A deeper look into the regional CBB vulnerability in different locations in Colombia indicates that the two largest coffee-producing departments (comparable to states), Antioquia and Huila, represent the worst and the best scenarios for CBB damages in field conditions (Figure 7). These findings are in concordance with the vulnerability conditions predicted by the temperature conditions (Table 2).

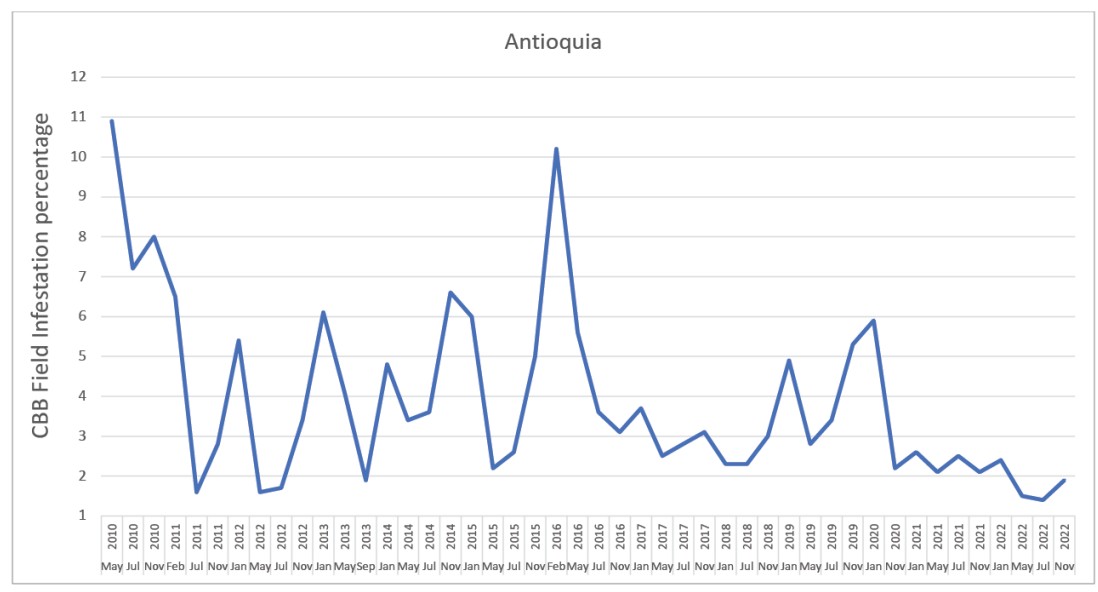

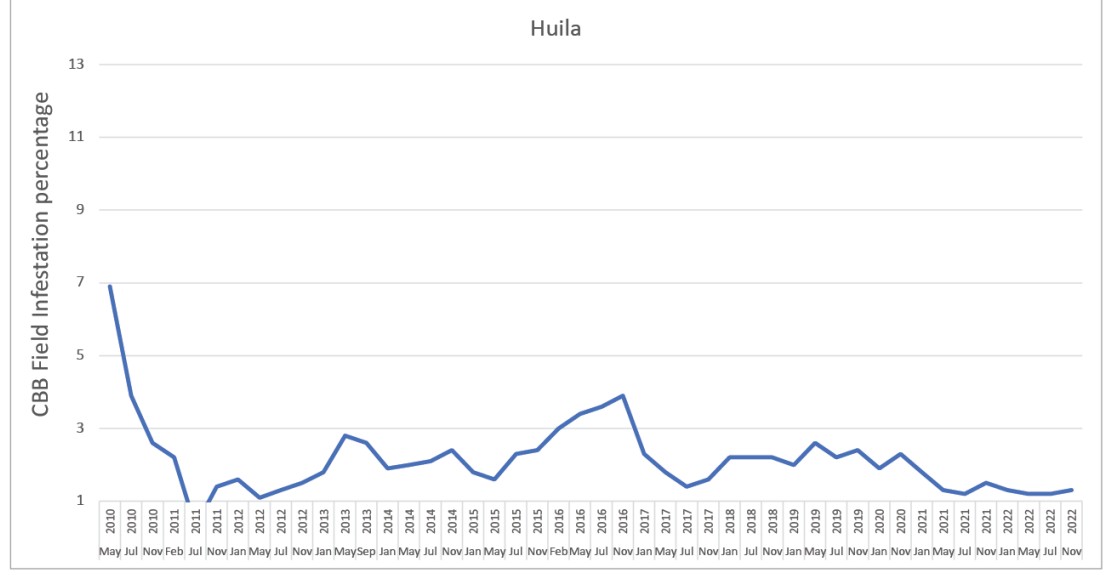

**Figure 7.** CBB field infestation levels in the two largest Colombian coffee-growing departments: Antioquia and Huila (comparable to states) (adapted and updated from Giraldo-Jaramillo et al. [5,6]).

**Table 2.** Percentage of the coffee-growing area of the two largest Colombian coffee-growing departments (comparable to states) vulnerable to the coffee borer depending on ENSO events [5,6].

| ENSO | Coffee Growing Area (%) | | | | | | | |
|---|---|---|---|---|---|---|---|---|
| **EVENT** | **Very Low** | | **Low** | | **Moderate** | | **High** | |
| **Locality** | **Antioquia** | **Huila** | **Antioquia** | **Huila** | **Antioquia** | **Huila** | **Antioquia** | **Huila** |
| Neutral | 9.1 | 7.8 | 39.0 | 49.9 | 50.9 | 42.1 | 1.0 | 0.2 |
| La Niña | 17.7 | 20.8 | 41.1 | 65.0 | 38.9 | 14.1 | 2.3 | 0.1 |
| El Niño | 4.0 | 2.1 | 28.7 | 57.0 | 55.3 | 39.0 | 12.0 | 1.9 |

Taking all these together, it is advisable to keep locally monitoring CBB in Colombia in order to keep this key coffee pest under threshold levels, especially during El Niño ENSO events.

New regulations concerning the use of chemicals in agriculture are limiting the use of certain insecticides, and the tendency is moving toward zero pesticides [7]. In this review, we provide the bases to establish an IPM program excluding chemical insecticides and focusing on new novel technologies that soon will appear as solutions for controlling CBB.

### 2.3. Economic Damage

The IPM for CBB in Colombia has allowed to keep the infestation levels under 3% nationwide. As a consequence, losses up to 180 million dollars per year have been avoided with this strategy. However, adding the cost of controlling the borer to the loss of 3% of infestation would add up to USD 66 million [8].

### 2.4. CBB Integrated Pest Management (IPM)

The European Green Deal emerges as an immediate action strategy to adopt the different challenges and commitments on climate change issues, with the main objective of providing a clear response to environmental problems and establishing an economic growth strategy based on the efficient use of the resources. In addition, by the year 2050, a complete reduction in emission of greenhouse gasses is expected. Given that, the objectives of the European Green Pact are of global priority and are framed in the following elements [7]:

1.　Higher level of climate ambition in the European Union, 2030 and 2050.
2.　Supply of clean, accessible, and secure energy.
3.　Mobilization of the industry in the direction of a clean and circular economy.
4.　Efficient use of energy and resources in construction and renovation.
5.　Accelerate the transition to sustainable and intelligent mobility.
6.　"From farm to table": a fair, healthy, and environmentally friendly food system.
7.　Preservation and restoration of ecosystems and biodiversity.
8.　Towards zero pollution in an environment without toxic substances

Regarding the "From farm to table" strategy, the objectives are to reduce—at least by 50% by 2030—the dependence on pesticides and antimicrobials, reduce excess fertilization, increase organic agriculture, and stop the loss of biodiversity. In addition, there are other binding aspects, such as the maximum limits of pesticide residues, for which the requirements will be greater and a series of agrochemicals would enter into a prohibited list, increasingly limiting their use for the control of agricultural insect pests. Consequently, it is important to ensure IPM in agriculture [7].

Since 1996, Cenicafé has established a strategy for the IPM of CBB [9], which involves aspects such as:

- Flowering registration to detect the critical period when the coffee fruit stages are most susceptible to be attacked by CBB.
- Evaluation of the percentage of CBB infestation in the field and the position of the insect in the coffee bean to identify the opportune moment for the use of insecticides (chemical or biological).
- Identification of aggregated CBB population areas to focalize control actions.
- Timely harvesting and removing unharvested coffee berries.
- Use of biological controllers such as entomopathogens and parasitoids.

As a result, when these practices are applied systematically, there is a decrease in the percentages of infestation of the CBB in the field and consequently in the commercial coffee parchment. Additionally, a greater volume of annual production has been accomplished. Moreover, it is possible to manage CBB and produce export-quality coffee with the IPM strategy [9]). Similarly, Constantino et al. [10] evaluated the control of CBB when collecting coffee berries from the ground with a cherry basket collector, demonstrating that more than 70% of the fallen fruits are removed. Additionally, this activity allowed maintaining the

infestation by CBB at levels lower than 3% during the main harvest, in comparison with the controls, which reached values higher than 7%. Additionally, in studies carried out by Benavides et al. [11], it was confirmed that the participation of cultural control, by means of opportune harvesting and removal of unharvested coffee berries, is the most important component in the control of the CBB and that the use of chemical insecticides as the only control alternative is ineffective.

The IPM strategy to control CBB was first validated in a coffee farm in the municipality of Chinchiná and allowed to produce 83% of the coffee parchment of the main harvest with quality type "Exportable Federation type" [9]. Briefly, the IPM strategy consisted of:

–    Characterization of coffee lots and identification of CBB aggregation areas.
–    Cultural control as a fundamental basis for the IPM program by means of timely harvesting and removing unharvested coffee berries twice a year after main harvesting ended.
–    Spraying chemical insecticides at the opportune moment and on the CBB aggregation areas solely during the critical period of the attack of the borer (after 120 days after main flowerings) and when both field infestation exceeds 2% and more than 50% of the borers were in coffee fruits entrance positions just boring the coffee berry [12].
–    Spraying *Beauveria bassiana* (Bals.-Criv.) Vuill. on CBB aggregation areas every other week during the critical period of the attack of the borer.

On the other hand, Vélez and Benavides [13] isolated and characterized the fungus *Beauveria bassiana* from samples of natural infected CBB infesting coffee berries, registering this fungi genus as promising in the biological control of the borer. Likewise, Jaramillo et al. [14] found that the applications of *B. bassiana* affect the oviposition of the insect by up to 87%, decreasing its progeny in infested coffee beans. Today, *B. bassiana* is used as an important biocide in the control of CBB in Colombia. Complementary, Cruz et al. [15] and Cárdenas et al. [16] evaluated the mortality of the CBB caused by seven strains of *B. bassiana* and their mixtures, finding up to 100% mortality in the laboratory, while in the field, mortality ranged between 53.3 and 66.6%. Additionally, Vera et al. [17] found that the application of the mixture of different strains of *B. bassiana* reduced the percentages of infestation between 30 to 50%, causing mortalities close to 40% and a reduction of between 55 and 75% of the borer population within newly infested fruits.

According to Góngora et al. [18], to achieve high efficacy of *B. bassiana* fungus, field applications should be at a concentration of $2 \times 10^{10}$ spores/L, using a formulation with a purity greater than or equal to 95%, a concentration larger than $1 \times 10^9$ spores/g, and with more than 90% of spores germinating during the first 24 h.

Additionally, to *B. bassiana*, there are other fungi that infect the CBB, such as *M. anisopliae*. Milner and Lutton [19] reported that this fungus is better adapted to soil conditions than *B. bassiana*, being widely used in pest control at the rhizosphere level.

Bustillo et al. [20] evaluated the effect of spraying *B. bassiana* strain Bb-9205 and *M. anisopliae* Ma-9236 to the soil on the CBB that emerges from fallen fruits. These works were expanded by those developed by Vera et al. [17]. Jaramillo et al. [14] validated the effect of the mixture of strains of *B. bassiana* and its combination with *M. anisopliae* on the CBB that emerges from coffee fruits left on the ground. In the field, all treatments reduced the infestation in the trees by 18% and 47%, compared to the control, obtaining maximum control with mixtures of strains of *B. bassian* and *M. anisopliae*.

Other controllers that have been evaluated are nematodes of the families Steirnernematidae and Heterorhabditidae for the control of individuals present in fruits fallen to the ground [21]. When *Steinernema colombiense* is mixed with entomopathogenic fungi such as *M. anisopliae* and *B. bassiana*, they could be capable of transporting conidia into the fruit and reducing the potential population of borers with the capacity to infest healthy fruits [21]. However, the results under field condition have been contradictory due to the high mortality of the nematodes.

Here, we present a general scheme (Figure 8) for accomplishing a sustainable IPM program to control CBB in the basis of real flowering events for the central coffee producing area of Colombia (Naranjal Cenicafe's Main Experimental Station located in Chinchiná,

Caldas, Colombia. Located at 04°59′ N, 75°39′ W, and an altitude of 1.400 m.a.s.l. Summarizing, the critical period for borer attack occurs after 120 days of the main and midterm flowering. During these periods, the coffee berry contains more than 20% dry matter, so CBB reproduction and development can complete inside the coffee beans [3].

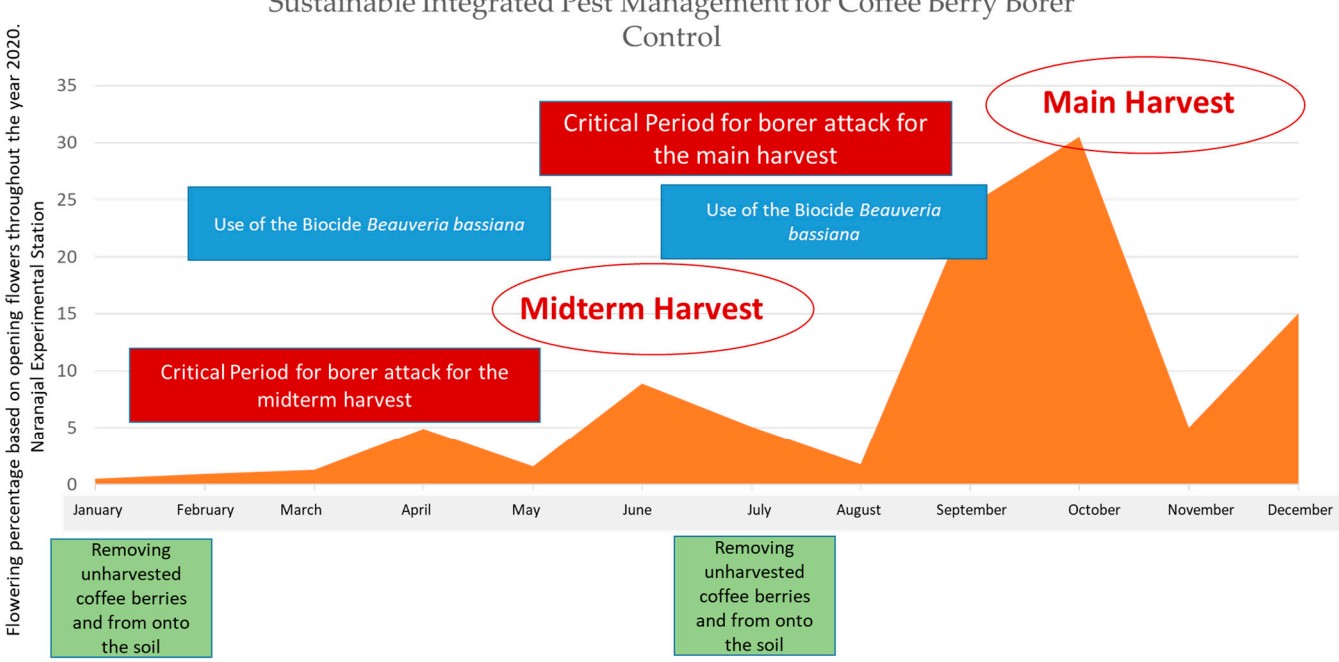

**Figure 8.** New sustainable integrated pest management for CBB.

During these periods, spraying the biocide *B. bassiana* could take place if the infestation levels in the field surpassed 2%. For this, it is recommended to count the total number of coffee berries per branch and those infested by CBB in 30 coffee trees randomly selected in a W path per hectare of coffee lot. Additionally, the percentage of alive CBB adults entering the coffee berry must be larger than 50%. For this, we take 100 coffee berries during the evaluation of the field infestation and register the number of CBB alive that are just boring the fruit. Therefore, we guarantee that the biocide would target the CBB adults at the right moment. During the midterm and main harvests, collecting ripe coffee fruits in a timely manner will cut the cycle of CBB; however, removing unharvested coffee berries and those left onto the soil should be performed twice a year as a way to clean up the coffee crops. This cultural control activity is responsible for most of the IPM program since it has been demonstrated to keep CBB infestation levels under the economic threshold level of 5% [10].

*2.5. CBB Control Prospective*

Cultural control, which consists of the periodic and timely collection of ripe coffee beans, is essential to controlling the CBB pest. This control strategy contributes to interrupting the communication between the coffee plant, specifically the fruit, and the insect in such a way that the ripe coffee fruits, which are the organs in the plant that emit the signals that most attract the insect, are removed. Recently, at Cenicafe, attracting substances for the CBB present in the coffee fruits and repellent substances for the CBB present in other plants have been identified and can be used as part of the insect control strategy. Initially, Castro et al. [22] identified CBB-repellent plants, such as *Emilia sonchifolia* (Asteraceae), and volatile insect repellents, such as β-caryophyllene [23]. With these plants and volatiles a push–pull agroecological strategy was proposed that consists of using repellent plants or volatile repellent devices inside or around the coffee crop and attracting volatile devices or attracting plants outside the plot during the critical period of the attack of the CBB with the main objective of reducing the population of borers within the coffee plantation [24].

Moreover, a large-scale (area-wide) biological control strategy with the use of CBB parasitoids has been recently tested in the field and has shown successful results in its application. In summary, the strategy consists of the release of the specific parasitoid predator *Prorops nasuta* (Waterston, 1923) (Hymenoptera: Bethylidae) before stumping coffee plantations in order to reduce dispersal rates of CBB and the release of the CBB adult specific parasitoid *Phymastichus coffea* (LaSalle, 1990) (Hymenoptera: Eulophidae) in nearby coffee plantations, where the coffee fruits are under formation in order to reduce colonization rates [25]. The combination of the different strategist depending on the stage of ripening of the fruits and the phenology of the coffee plant allowed appropriate control of the pest.

### 3. *Monalonion velezangeli* Carvalho and Costa, 1988 (Hemiptera: Miridae)

The species *Monalonion velezangeli* is known as the avocado bug or coffee bug. It was registered for the first time in Colombia in 1984 in fruits of avocado *Persea gratissima* Gaertn. (Lauraceae) and was described by Carvalho and Costa in the year 1988 [26,27]. Later, in 1998, it was reported causing lesions in coffee *Coffea arabica* L. (Rubiaceae) and only until 2008 it was identified as the causal agent of the disturbance known as coffee "chamusquina" [28]. Nymphs and adults feed mainly on the stems, leaves, and young shoots of the plant, causing the burning and death of these organs [29].

#### 3.1. Insect Biology

*M. velezangeli* is hemimetabolous and goes through the stages of egg, nymph, and adult. The duration of the life cycle is variable according to the host; that is, in coffee (*Coffea arabica* Rubiaceae) at $20 \pm 2\,°C$ and a relative humidity of $78 \pm 10\%$, the duration from egg to adult takes $55.93 \pm 2.4$ days [30] (Figure 9), while in avocado (*Persea americana* Lauraceae), under conditions of $21.6\,°C$ and $78\%$ R.H., the duration from egg–adult takes $39.6$ days [29]. In the case of cocoa (*Theobroma cacao* L. Malvaceae), the egg–nymph duration takes $38.75 \pm 2.15$ days [30]. For the three cases, the longest duration was in coffee.

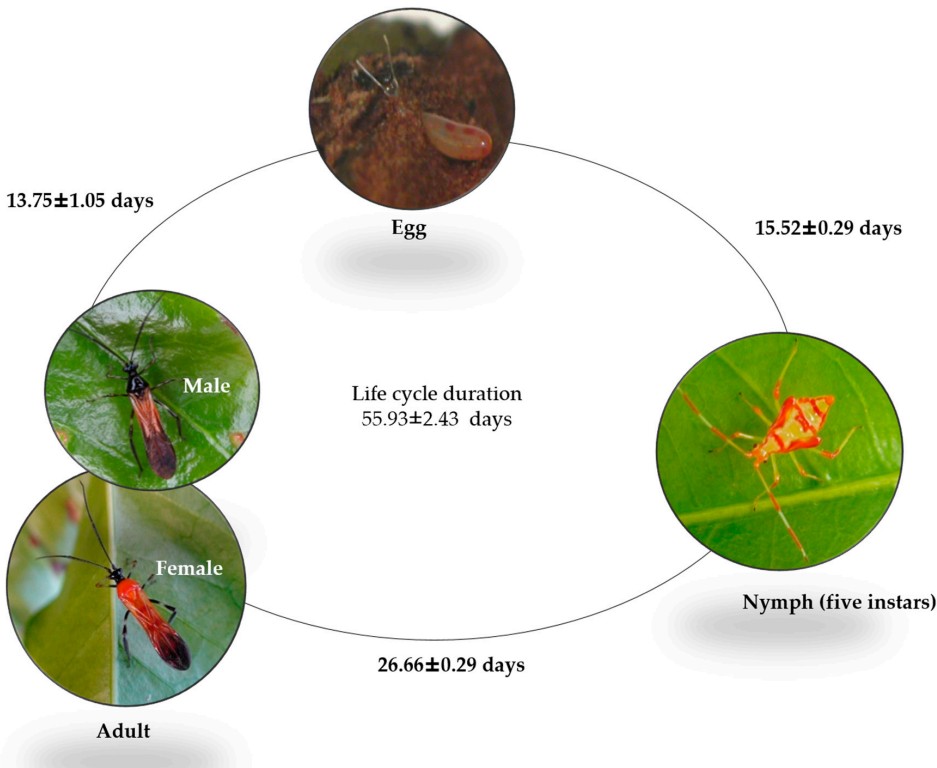

**Figure 9.** *Monalonion velezangeli* life cycle in coffee.

The reproductive rate also depends on the host; in coffee, the average number of eggs per female is 1.42, which is why it is considered low compared to the primary hosts, cocoa, in which the average number of eggs/female is 13.2. In the case of avocado, the average number is 13.05 ± 1.24 eggs/female [30]. This indicates that it prefers plants other than coffee for feeding and reproduction, but in the absence of those, it attacks coffee.

### 3.2. Insect Damage

The species *M. velezangeli* is a sucking insect that sucks the sap in the stems, leaves, flowers and fruits reducing the growth and production of various host plants [31,32] (Figure 10).

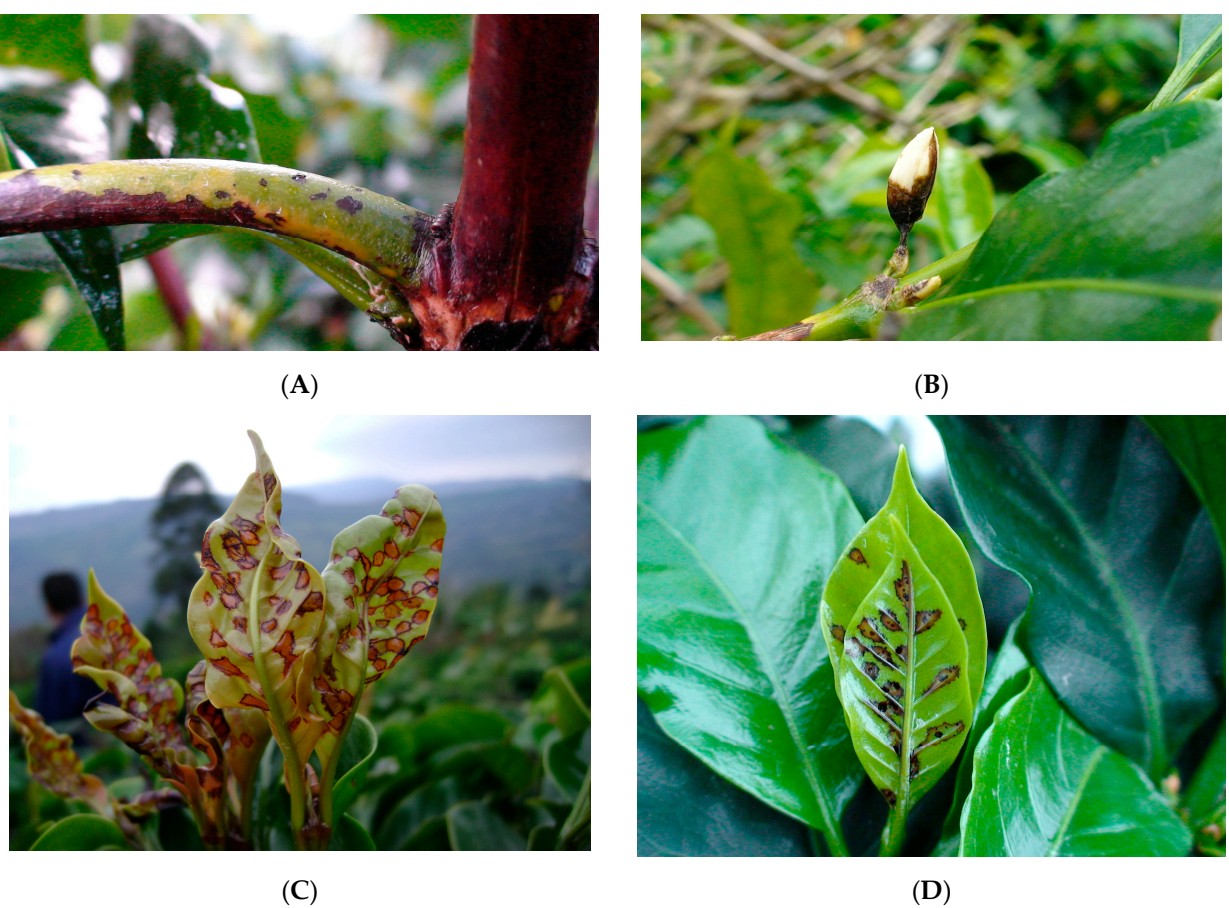

(**A**)

(**B**)

(**C**)

(**D**)

**Figure 10.** *Monalonion velzangeli* damage in coffee. (**A**) Stems; (**B**) flowers; (**C**) leaves; (**D**) young shoots.

In coffee, *M. velezangeli* causes damage, altering the plant's development in coffee-growing areas in southern Colombia located at altitudes above 1550 m.a.s.l., with luminosity below 1400 h/year, temperatures below 20 °C, and relative humidity above 80% [33]. The insect populations increase and cause the greatest damage during the rainy periods and during "La Niña" climatic events.

### 3.3. Economic Damage

As it is a pest of recent appearance in Colombian coffee growing, the economic injury level (EIL) and the economic threshold (ET) have not been determined; however, production losses were estimated to range from 37.8 to 50% per tree and appear 15 months after the attack (Cenicafe, Informe Anual de Labores 2020, pg. 60 https://doi.org/10.38141/10783/2020).

### 3.4. Control Strategies

Management strategies for this species are focused on natural control, biological control, and cultural control.

*M. velezangeli* populations are naturally regulated by Reduviidae (Hemiptera), generalist predators which need to consume enough prey to complete their development and longevity; therefore, the presence of these predators is important in coffee plantations to reduce the populations of this pest. The most common species present in coffee crops are: *Arilus gallus* (Stål, 1872); *Castolus lineatus* (Maldonado, 1976); *Repipta* sp., and *Zelus vespiformis* (Hart, 1987) [28,29].

Regarding biological control agents, a virulent strain of the entomopathogenic fungus *Beauveria bassiana* previously isolated from *Monalonion* sp. (Hemiptera: Miridae) was evaluated under laboratory and field conditions. The results showed that applying a concentration of $4 \times 10^{10}$ spores/L. under laboratory conditions caused mortality of 80%. These results were validated in the field and demonstrated that fungus applications allow longer pest control times when compared to generalized applications of chemical insecticides [34].

Cultural control consists of making the environment less favorable for the development of the pest in the field. For this, it is recommended to remove the climbing vine *Cissus verticillata* (L.) Nicolson and C.E. Jarvis (Vitaceae) from the coffee plantations. *Monalonion velezangeli* reproduces on this vine, and when the nymphs emerge, they also attack coffee [29]. Additionally, it is recommended to keep primary hosts of the pest, such as *Psidium guajava* L. (Myrtaceae) and avocado *Persea americana* (Lauraceae), within coffee lots; in this way, the insect is prevented from passing through and causing damage to the coffee trees. Additionally, it is recommended that integrated management of weeds (IMW) be carried out to promote the permanence and reproduction of *M. velezangeli*'s natural enemies; it has been found that in the absence of weeds, attacks by *M. velezangeli* increase in coffee crops [28,29].

### 3.5. Monalonion Velezangeli and the Relationship with Weather

Understanding the relationship between environmental factors and the pest population can not only help anticipate economic losses in a crop but can also contribute to avoiding them through the implementation of timely control measures [35].

In order to relate the attacks of the coffee chamusquina bug with the climate variables, the average temperatures of the day and night were analyzed at the altitudes at which the greatest damage occurred. The loss occurs at altitudes above 1550 m.a.s.l.

It was found that the climatic variable: relative humidity at night, when the relative humidity decreases by up to 75% on average, was associated with the appearance and increase in coffee shoots affected by chamusquina. Additionally, it was possible to determine the relative humidity value from which, 15 days later, the attacks of the insect would begin in the field. This is how the increase in the average temperature from 19 °C to 22 °C before the attack, the decrease in relative humidity to less than 55% during the day, and the progressive increase in relative humidity up to 95% in the night over a period of 30 days favors the development of and increase in populations in the field [29] (Figure 11).

Thus, the changes in nocturnal relative humidity would be the early warning event that would allow decisions to be made before the attack of the pest. At this time, it is necessary to start the monitoring and timely control of chamusquina in vulnerable regions with high altitudes. More studies that seek to relate this variable at a microclimatic scale to the biology and reproduction of *M. velezangeli* on coffee should be made, and later, control strategies that include modification of the environment to affect the biological development of the insect should be implemented [29].

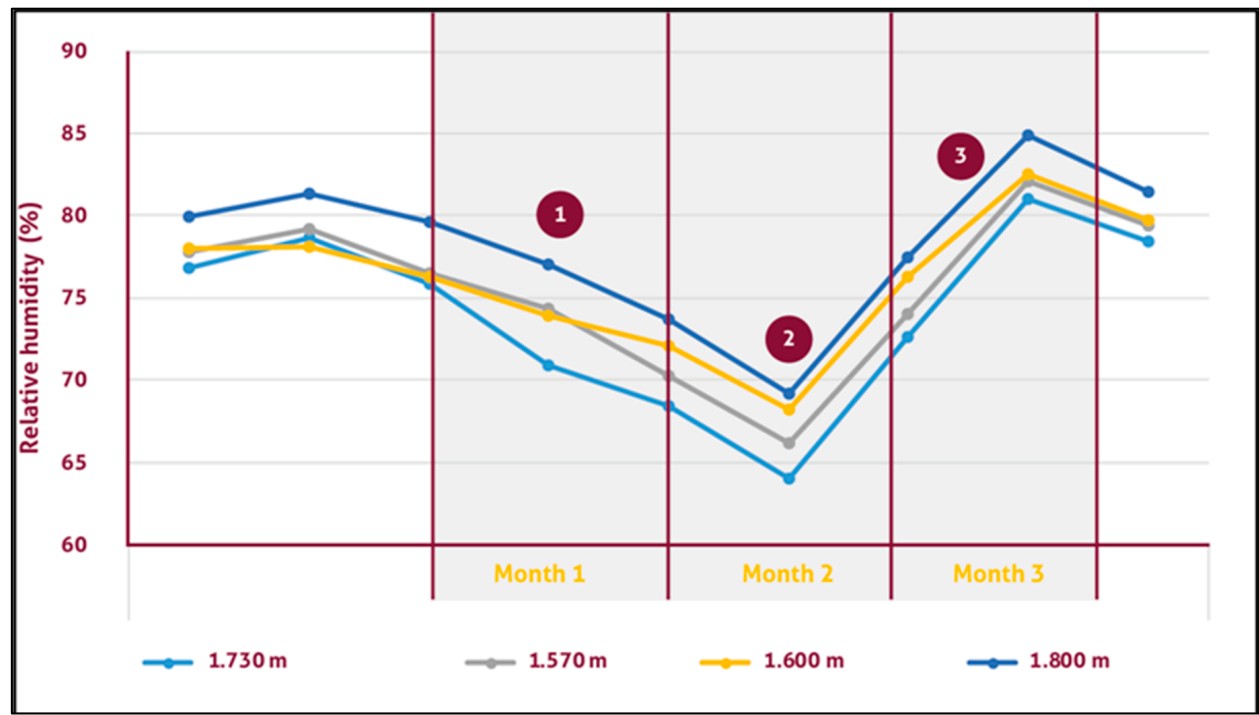

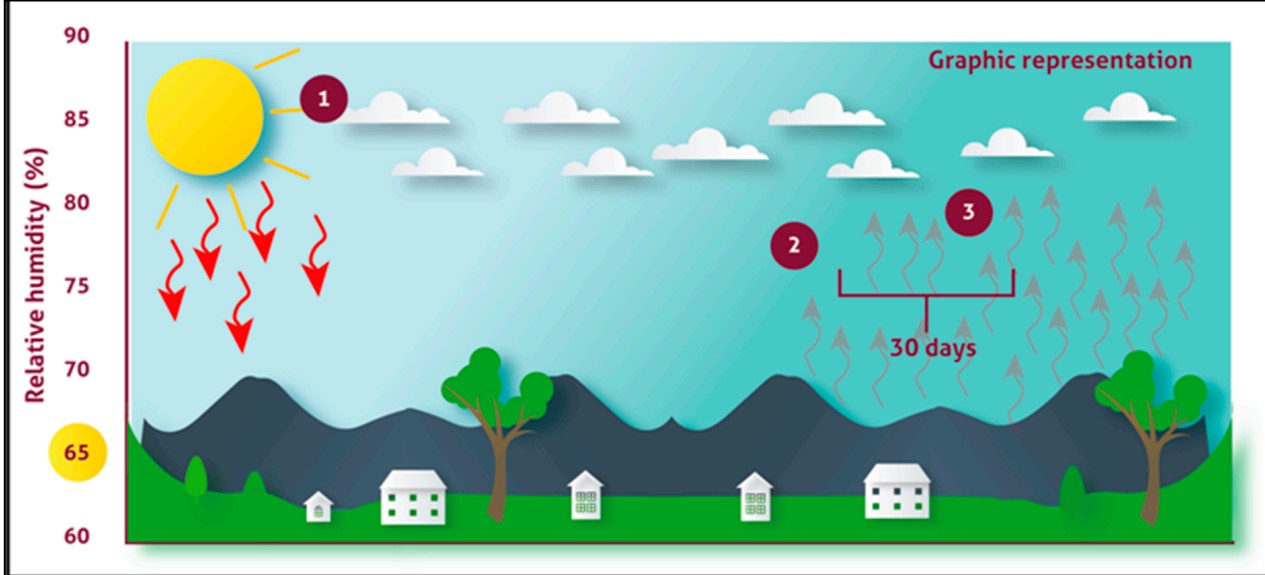

**Figure 11.** Explanatory diagram of the behavior of the climate in the regions where there is chamusquina infesting coffee. (1) Increase in the average temperature before the attack ≥19 and <22 °C. (2). Decrease in relative humidity to less than 55% during the day. (3). Progressive increase in relative humidity up to 95% at night over a period of 30 days. (m = m.a.s.l)

To face climate change, it is necessary to implement integrated control strategies that include natural control, biological control and cultural control for this pest, as indicated in Figure 12, in order to achieve sustainable management of coffee cultivation.

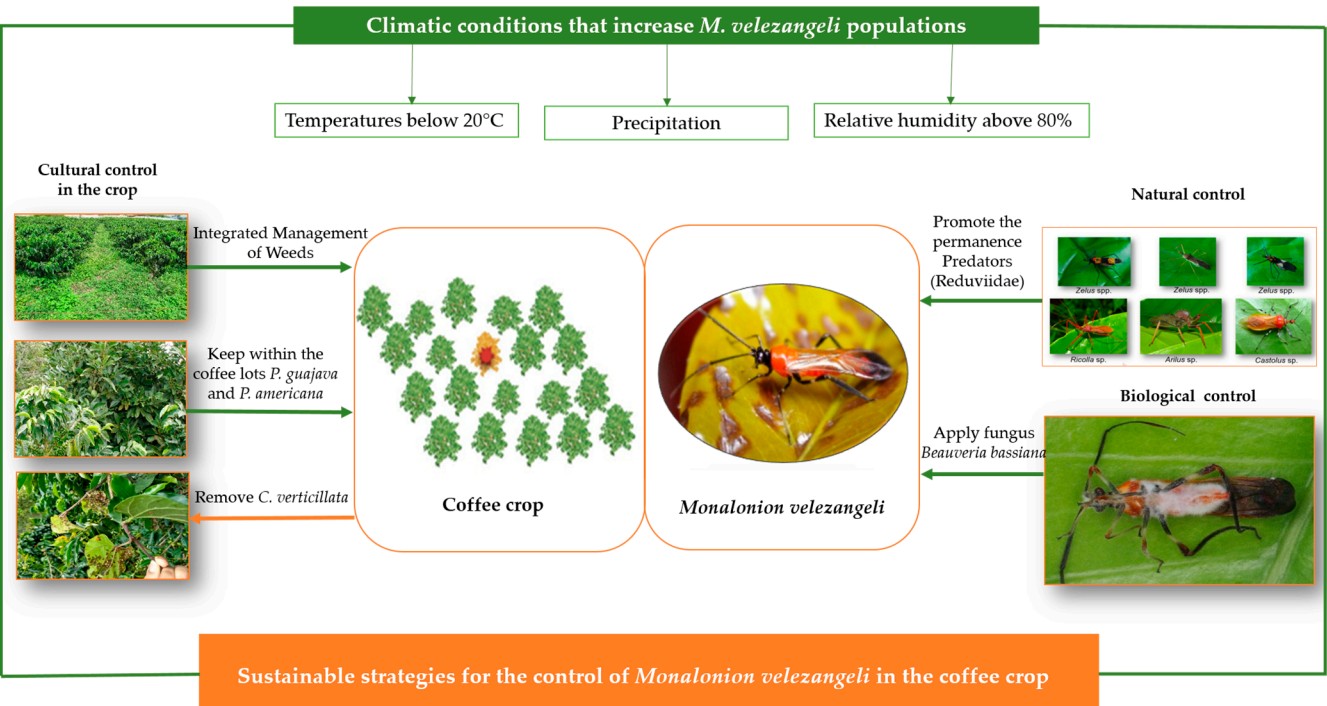

**Figure 12.** Approach to the sustainable management of *Monalonion velezangeli* in the coffee crop. Interactions between the main components for control.

## 4. Coffee Root Mealybugs

Coffee root mealybugs are insects with phytophagous habits of the order Hemiptera, infraorder Coccomorpha. They use various plants as hosts, including crops of agricultural, forestry, and horticultural importance, and they also feed on wild plants [36,37].

In Colombia, 65 species of mealybugs associated with coffee roots are registered, of which 20 are new records in coffee and 15 new records for Colombia [38,39].

The species *Puto barberi* (Cockerell, 1895) (Hemiptera: Putoidae) stands out because of its abundance and prevalence, as do *Dysmiccocus* complejo *texensis* (Tinsley, 1900), *D. brevipes* (Cockerell, 1893), *D. neobrevipes* (Beardsley, 1959), *Dysmiccocus* sp., and *Pseudococcus elisae* (Borchsenius, 1947) (Hemiptera: Pseudococcidae), because they are associated with basidiomycete fungi, which encyst coffee roots and cause plant mortality. *Neochavesia caldasiae* (Balachowsky, 1957) (Hemiptera: Rhizoecidae) and *Toumeyella coffea* Kondo, 2013 (Hemiptera: Coccidae) also cause plant mortality, and ultimately, *Geococcus coffeae* (Green, 1933) and *Rhizoecus colombiensis* (Hambleton, 1946) (Hemiptera: Rhizoecidae) are also abundant [38,40]. The cryptic habit of these species hinders timely diagnosis and the effectiveness of control agents [41].

### 4.1. Insect Biology of Puto barberi and Pseudococcus elisae

*Puto barberi* presents incomplete metamorphosis; it goes through the stages of egg, nymph (immature stages), and adult. Under controlled conditions (25 $\pm$ 2 °C and 70$\pm$ relative humidity), the duration from nymph to adult takes 141 $\pm$ 0.99 days. The duration of the egg is not known since this insect is ovoviviparous and the eggs remain inside the body of the female until the embryo is fully developed [42].

The *P. elisae* life cycle was carried out under controlled conditions (25 $\pm$ 2 °C, 70–80% RH and photoperiod of 12:12 h). The duration of the females' transition from egg to adult takes 98.1 $\pm$ 1.3 days; three nymphal instars were recorded, with a duration of 61.4 $\pm$ 0.3 days. The nymphal stage lasted 9.3 times longer than the egg stage. Females go through the stages of egg, nymph I, nymph II, nymph III, and adult. In the case of males, the transition from egg to adult takes 68 $\pm$ 0.4 days, and it goes through the stages

of egg, nymph I, nymph II, prepupa, pupa, and adult; the longest duration occurred in the nymph stage, with $31.2 \pm 0.5$ days, and the shortest duration was the egg stage, with $6.6 \pm 0.3$ days [43] (Figure 13).

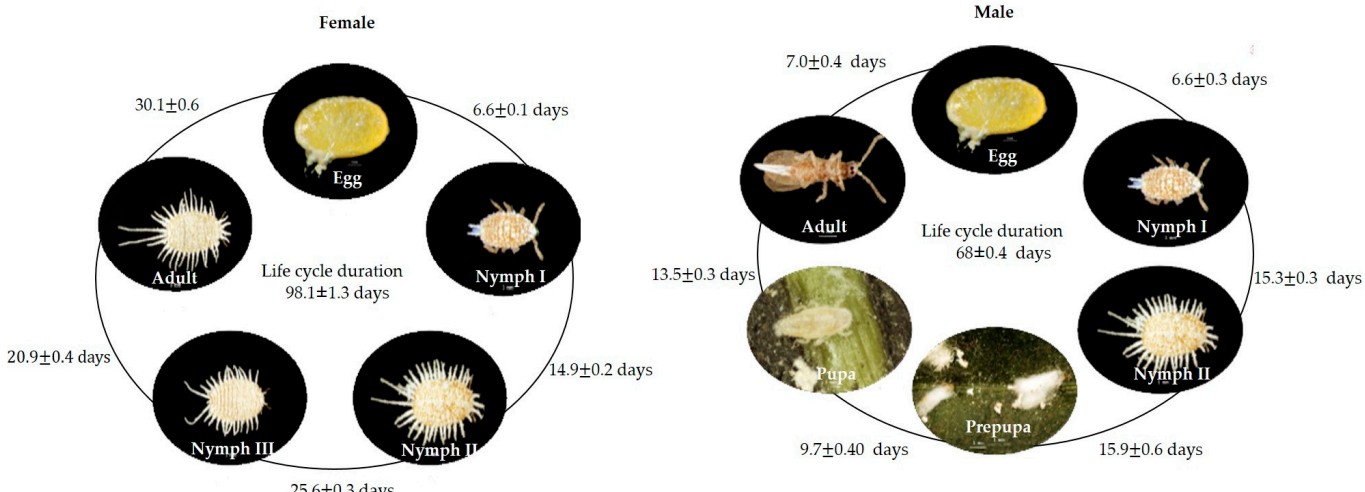

**Figure 13.** *Pseudococcus elisae* life cycle in coffee.

### 4.2. Insect Damage

In coffee *Coffea arabica* L. (Rubiaceae), the insect causes damage to the plant with the biting–sucking mouthparts, sucking the sap, and as a consequence causes chlorosis, defoliation, and in some cases death, especially in coffee plantations less than two years old [43] (Figure 14).

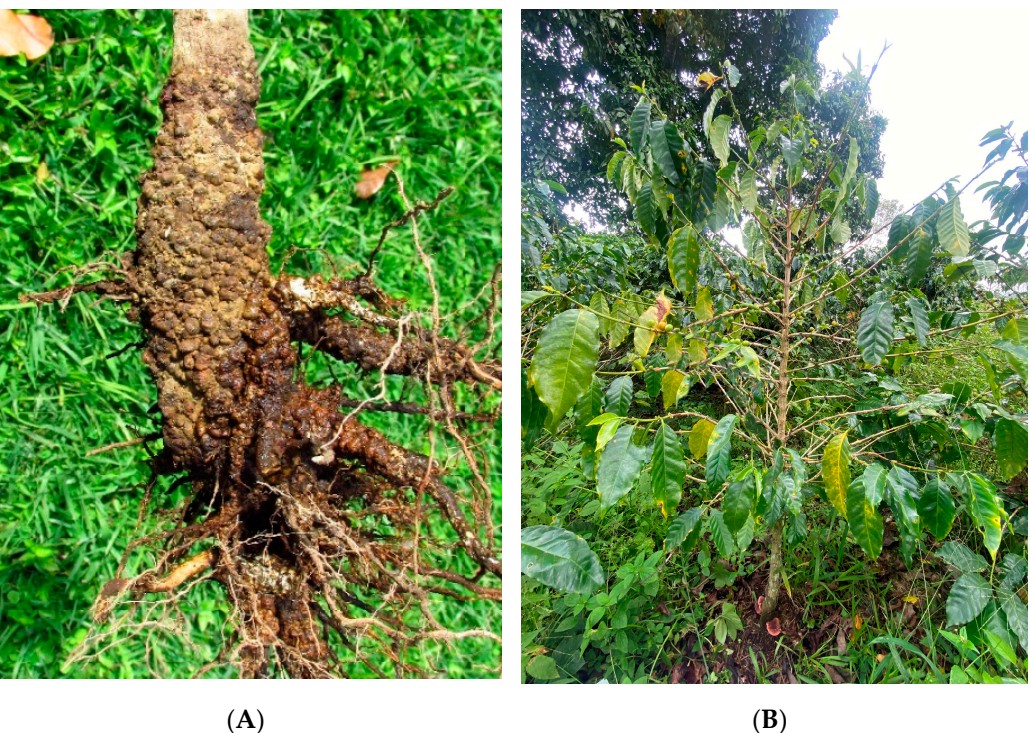

(**A**)                    (**B**)

**Figure 14.** Mealybug damage. (**A**) Coffee root with cysts produced by Pseudococcidae. (**B**) Coffee tree with chlorosis and defoliation.

### 4.3. Economic Damage

Mealybug populations in coffee increase and cause the greatest damage during rainy periods and in "La Niña" weather events; while in dry periods and "El Niño" climate

scenarios, individuals go deeper and colonize the terminal part of plant roots [43]. In coffee cultivation, they are distributed in different altitudinal ranges, Caballero et al. (2019) [38] reported that *P. barberi* and *P. elisae* occur between 700 to 1900 m.a.s.l., while *Dysmicoccus texensis* complex, *G. coffeae*, and *R. colombiensis* occur between 1000 and 2200 m.a.s.l.

Recent research in Colombia estimated the losses in production as a result of the damage that some species of mealybugs cause to coffee roots; it was found that plant mortality caused by *N. caldasiae* represented a 25.1% and 68% of reduction in production. For Pseudococcidae mealybug species that are associated with basidiomycete fungi, the percentage of plant mortality was 14%, and the decrease in production was 25.1% (Cenicafe, Informe Anual de Labores 2023, pg. 40 https://doi.org/10.38141/10783/2023).

### 4.4. Control Strategy

**Natural control:** In relation to the natural enemies of mealybugs present in coffee cultivation in Brazil, Rodrígues et al. [44] reported the generalist predators as the main mortality factors of *Planococcus citri* (Risso, 1813), such as *Harmonia axyridis* (Pallas, 1773), *Cycloneda sanguinea* (Linnaeus, 1763), *Azya luteipes* (Mulsant, 1850), *Diomus seminulus* (Mulsant, 1850), *Diomus sennen* (Gordon, 1999), *Cryptolaemus montrouzieri* (Mulsant, 1853), *Cyra loricata* (Mulsant, 1850), *Hyperaspis festiva* Mulsant, 1850 (Coleoptera: Coccinellidae); *Chrysoperla genanigra* (de Freitas, 2003), *Chrysoperla externa* (Hagen, 1861) (Neuroptera: Chrysopidae), *Allograpta* sp., *Ocyptamus* sp. (Diptera: Syrphidae), and *Condylostylus* sp. (Diptera: Dolichopodidae). In India, Balakrishnan et al. [45] reported as natural enemies of *Ferrisia virgata* (Cockerell, 1893) (Hemiptera: Pseudococcidae) the parasitoids *Aenasius advena* (Compere, 1937), *Anagyrus qadrii* (Hayat, Alam, and Agarwal, 1975) and *Blepyrus insularis* (Cameron, 1886) (Hymenoptera: Encyrtidae). Additionally, three predatory species of a genus close to *Scymnus* (Coleoptera: Coccinellidae) were reported. Meanwhile, for Colombia, Gil et al. [46] reported *Aenasius caeruleus* (Brues, 1910), *Aenasius bolowi* (Mercet, 1947), *Aenasius tachigaliae* (Brues, 1922), *Aenasius* af. *mitchellae* (Noyes and Ren, 1995), *Zarhopalus putophilus* (Bennett, 1957), *Hambletonia pseudococcina* (Compere, 1936), *Hambletonia* sp., *Prochiloneurus* af. *dactylopii* (Howard, 1885), *Leptomastix* sp., *Cicoencyrtus* sp. (Hymenoptera: Encyrtidae), five adult cecidomids (Diptera: Cecidomyiidae), two species of coccinellids (Coleoptera: Coccinellidae), and an antagonistic fungus, *Trichoderma* sp.

As for the pathogens, according to Shylesja and Mani [47], only entomopathogenic fungi are reported as causing natural infection in mealybugs. Moore [48] reported 13 species of fungi recorded in different countries, including the fungus *Metarhizium anisopliae* (Metschn) (Sorokīn, 1883), which exercises control over mealybugs such as *D. brevipes*, *Ferrisia virgata* (Cockerell, 1893), *Planococcus citri* (Risso, 1813), *P. lilacinus* (Cockerell, 1905), and *Planococcus* sp. (Hemiptera: Pseudococcidae).

**Biological control.** *Metarhizium anisopliae* Ma 9236 and *M. robertsii* strains were evaluated in the laboratory at concentrations of $1 \times 10^7$ conidia/mL, with mortalities between 80% and 84% on *P. barberi*. They were also evaluated in coffee seedlings infested with the pest in concentrations of ($2 \times 10^{10}$ conidia/L). *M. robertsii* fungus causes mortalities similar to those caused by azadirachtin 6% water soluble powder (3 g/L) and chlorpyrifos 75% WG (3 g/L) and can protect the roots of the plant from this pest, so it becomes a good candidate for further studies under commercial nursery conditions and in the field [49].

**Cultural control** focuses mainly on avoiding the spread of the pest, timely detection, and conservation of natural enemies. To avoid dispersal, it is recommended to have healthy seedlings since they constitute the main source of dispersal of mealybugs; if a single plant has a presence, when taken to the field, it becomes a source of dispersion. For timely detection, the planting of indicator plants in the streets of the lots allows periodic monitoring of these. It is recommended to sow at least 360 indicator plants/ha and to sample 30 the first 12 months after the crop has been established. To carry out IMW with this practice, the permanence of natural enemies is encouraged; additionally, since mealybugs are polyphagous and stay in some weed species, if they are eliminated, they are passed to coffee [50].

*4.5. Ecological Associations*

**Association with ants:** There is a very close relationship between the development of root mealybugs and the activities of various ant species within the coffee plantation. The mealybugs cannot digest the sap they take from the plants; therefore, they excrete it through the anus as a sugary substance, on which the ants feed. To their advantage, the ants transport the mealybugs and protect them from natural enemies. In studies carried out in Colombia, 19 genera of ants associated with mealybugs on coffee roots were recorded, namely: *Solenopsis, Pheidole, Brachymyrmex, Tranopelta, Wasmania, Acropyga, Hypoponera, Prionopelta, Crematogaster, Linepithema, Odontomachus, Paratrechina, Cyphomyrmes, Monomorium, Heteroponera, Strumigenys, Carebara, Mycocepuros,* and *Typhlamyrmex* [51].

**Association with basidiomycete fungi:** Some species of Pseudococcidae are associated with basidiomycete fungi, which, in their initial stage, form a corky structure similar to galls or nodules, covering both the main root and the secondary roots of the plants, affecting absorption of water and nutrients and causing chlorosis and, in severe cases, death [35]. The fungus, the insect (mealybug), and the plant form a unique tripartite nutritional relationship in which the plant roots are partially or totally covered by fungal fruiting bodies that produce cysts in which the cavity walls are formed by hyphae of fungi instead of plant tissues, with root mealybugs inside [52]. In coffee cultivation, five species of fungi associated with four species of Pseudococcidae were identified: *Phlebopus beniensis* associates with *Pseudococcus elisae, Dysmicoccus* complejo *texensis, Dysmicoccus brevipes,* and *Pseudococcus* nr. *sociabilis; Pseudolaccaria pachyphylla* associates with *Dysmicoccus* complejo *texensis,* and *D. brevipes; Phlebopus portentosus* associates with *Dysmicoccus* complejo *texensis;* and the species *Xerophorus olivascens* and *Boletinellus rompelii* associate with *Pseudococcidae* spp. [43].

**Mealybugs and their relationship with the weather**

Abiotic factors play an important role in the development of mealybugs. Temperature and relative humidity (RH) are the main ecological factors that positively or negatively affect mealybugs and their natural enemies. Dry conditions as an effect of high temperatures favor reproduction and the number of annual generations [36]; however, mealybugs are soft-bodied insects which can lose moisture and die. In coffee, rain favors the reproduction of basidiomycete fungi associated with mealybugs, since during rainy periods, basidiocarps (fruiting bodies) are observed at the base of the trees [43].

**Sustainable integrated management of root mealybugs**

The previous sections described the tools and strategies available to reduce populations of root mealybugs in coffee and prevent spread and infestation.

Some biological control practices with the *Metarhizium robersii* fungus provided promising results [49], and there are several species of natural enemies that exercise natural control in the field. Therefore, exploration of an alternative of biological control via augmentation is recommended.

In general, control tools applied as a stand-alone treatment may not be completely effective for heavy infestations of coffee root mealybugs. Therefore, management requires a holistic approach, and rational strategies must be implemented to avoid the establishment of mealybugs, such as planting healthy seedlings. This is essential to avoiding infestations; additionally, the permanent monitoring of the indicator plants allows timely determination of their presence. In addition, the IMW and the conservation of the vegetation around the coffee plots enhances the abundance and permanence of natural enemies. The cultural practices described above aim to prevent the accumulation of mealybug populations at a harmful density, but they are not effective against high infestations, so management must be preventive, avoiding the spread and colonization of the pest in regions in which it has not yet been found.

Finally, advancement in the knowledge of the relationship of climate to the phenology of the crop, the biology of the pest, and its dispersal will contribute significantly to a sustainable management strategy for coffee root mealybugs.

## 5. Coffee Leaf Miners *Leucoptera coffeella* (Guerin-Meneville) (Lepidoptera: Lyonetiidae)

There are four species of coffee leafminers in the world and *Leucoptera coffeella* (Guerin-Meneville) (Lepidoptera: Lyonetiidae) is a monophagous species adapted to the genus *Coffea* and widely distributed in the neotropical region, where it is found attacking coffee crops across Central and South America. In Africa, other species of coffee leaf miners have been reported to be present in coffee plantations, such as *L. meyricki* (Ghesquiere, 1940), *L. comma* (Chesquiere, 1940), and *L. caffeina* (Washbourn, 1940) [53].

### 5.1. Insect Biology

Adults are nocturnal moths. Their life cycle is in Figure 15. A female can deposit between three and seven eggs on the underside of the leaves. In her short life, which lasts two to three weeks, she lays approximately 70 eggs. The larva reaches a length of 4 mm and is creamy white with constricted, screw-shaped body rings. The larva emerges from the bottom of the egg and begins to consume the epidermis of the leaf, penetrating the mesophyll of the tissue, where it forms irregular galleries inside the leaves. The larvae pupate on the underside of the leaves, covered with white silk threads in the shape of an "X" [34].

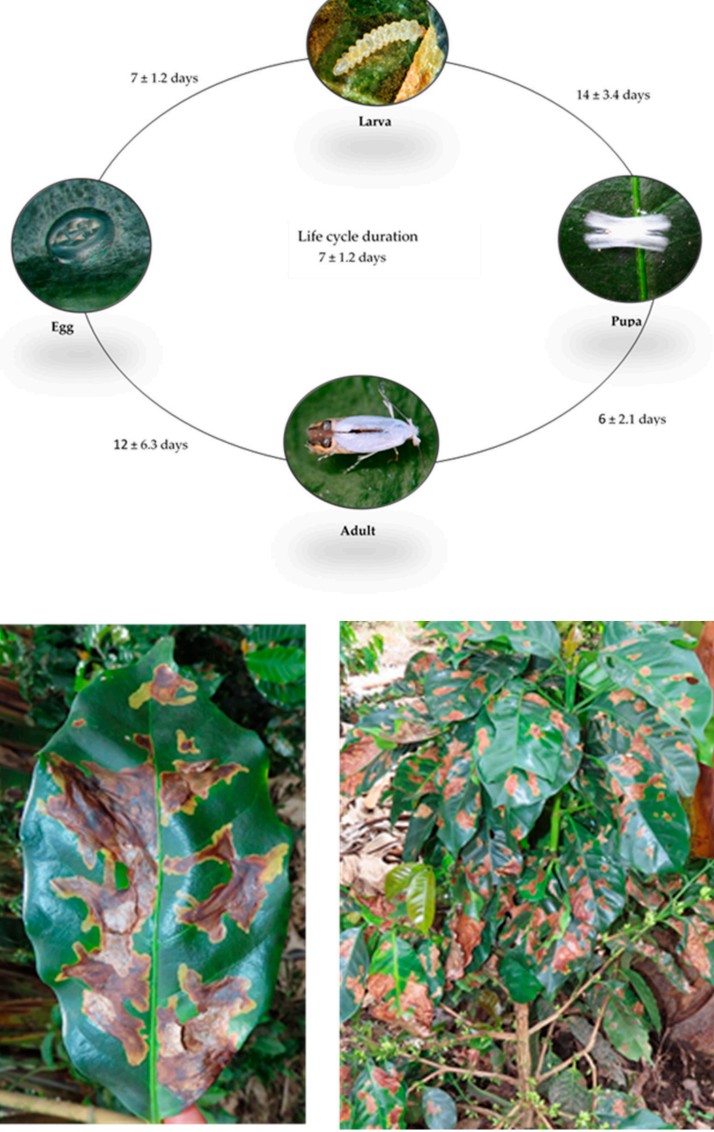

**Figure 15.** *Leucoptera coffeella*, life cycle and damage in coffee.

### 5.2. Insect Damage

The damage is caused by the larva when it begins to consume the epidermis of the leaf, penetrating the mesophyll of the tissue, where it forms irregular galleries inside the leaves. These lesions become necrotic and dry, turning brown. The damages caused by the leaf miner in coffee can be more critical if they coincide with high levels of incidence and defoliation with flowering periods, prolonged periods of drought, deficient fertilization, and attack by other pests [54]. According to Souza et al. [55] high defoliation can affect the formation of flower buds and consequently affect fruit production. Nantes and Parra [56], in a study in Brazil, reported that defoliation of 25%, 50%, and 75% resulted in coffee production losses of 9.1%, 23.53%, and 87.24%, respectively. Paliz and Mendoza [57] indicate that when leaf miner attacks coincide with flowering times, there are yield losses in fruit production that can exceed 50%. However, it has been noted that plantations with good fertilization are less prone to defoliation.

*L. coffeella* is distributed in coffee plantations located below 1300 m above sea level in low areas in conditions of relative humidity between 75 and 85% and temperatures between 22 and 25 °C. It affects coffee plantations of all ages both in full sun exposure and in those with regulated shading [54]. *L. coffeella* is a seasonal species with a higher prevalence in summer periods and during El Niño weather events, where strong attacks have been seen [58]. The most important factor in the population dynamics of leaf miner is the temperature—for each degree that it increases, it is possible to produce one more insect generation per year. If the average temperature is 18 °C, it is possible to obtain 6 generations per year, and with a temperature of 22 °C, 10 generations are obtained. Therefore, with the increase in temperature, the levels of the pest increase, and consequently, so does the damage to the coffee plantation [59,60].

### 5.3. Economic Damage

According to Souza et al. [55] high defoliation can affect the formation of flower buds and consequently affect fruit production. Nantes and Parra [56] reported that defoliation of 25%, 50%, and 75% resulted in coffee production losses of 9.1%, 23.53%, and 87.24%, respectively. Paliz and Mendoza [57] indicate that when leaf miner attacks coincide with flowering times, there are yield losses in fruit production that can exceed 50%. However, it has been noted that plantations with good fertilization are less prone to defoliation.

### 5.4. Strategies for Integrated Management Control of the Coffee Leaf Miner

The management of the coffee leaf miner should include a set of control practices:

**Biological control by conservation** is based on the modification of the environment or existing agronomic practices in a crop to protect and increase specific natural enemies or other organisms in order to reduce the effect of pests. Mainly, biological control via conservation differs from other biological control strategies in that it does not imply the release of natural enemies, but instead, it seeks to establish controllers through the application of certain methods in a natural environment in which the crop is developed [61]. The application of these methods does not by itself exercise the biological control of pests but promotes the abundance and diversity of natural enemies already present in the agroecosystem.

*L. coffeella* presents a large number of primary natural enemies of neotropical origin; 55 species that coexist with the coffee leaf miner, of which 43 species of parasitoids and 12 species of predators have been reported [62]. In Colombia, 15 species of parasitoids and 6 species of predators are found naturally controlling leaf miner populations [63,64]. With parasitism percentages above 60%, leaf miner populations remain below the level of economic damage. In order to maintain a food substrate for the parasitoids and predators, it is recommended to carry out selective control of weeds so that the soil maintains coverage of nectar and honey plants.

It has been possible to demonstrate that new coffee plantations during the summer are more susceptible to leaf miner attack when noble weeds are completely eliminated from the plots through the use of herbicides applied in a general manner (Figure 16). This destructive

practice of noble weed elimination makes the leaf miner's parasitoids and predators to disappear from the coffee plantation because the weeds produce flowers, nectar, and refuge for beneficial fauna adults, causing the migration of the natural enemies out of the crop. As the leaf miner does not have its natural enemies present, the plague populations increase considerably. However, when manual weeding is performed in the coffee tree plate and noble coverage is left in between the lines of the coffee plantation, the ecological balance is restored and the leaf miner populations decrease, maintaining the populations of the pest under natural control [64].

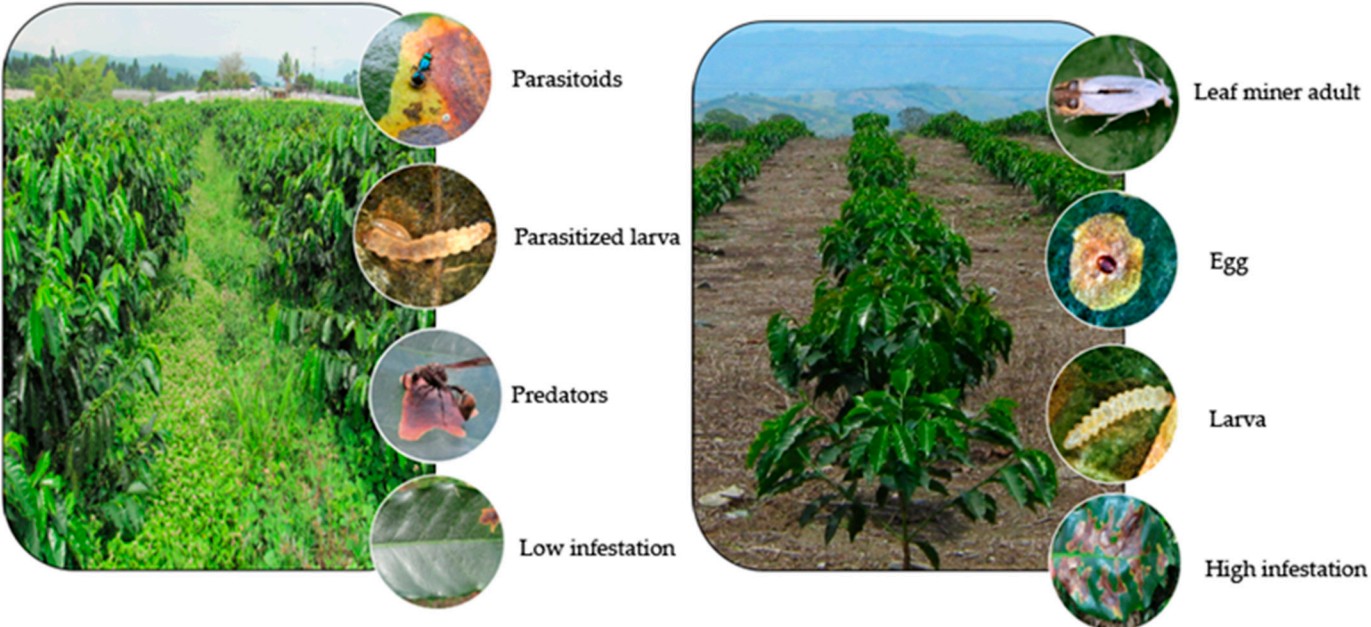

**Figure 16.** Biological control via conservation. The maintenance of noble weeds in coffee plantations favors the increase of natural enemies that regulate the populations of *Lecoptera coffeella* in contrast to crops without noble weeds, for which infestation levels increase.

**Ethological control.** The main form of communication between insects is through odors. The adult female leaf miner is capable of notifying the male through chemical signals when it is ready for copulation via the emission of a sexual pheromone. Francke et al. [65] identified the main components of the molecule of this pheromone. Once the chemical molecule was known, it was possible to synthesize it and produce it on a massive scale to use it as bait in traps to monitor the pest or to use it in the mating confusion technique. This technique consists of saturating the environment with the pheromone so that the males cannot distinguish the signals sent by the females. The use of traps for adult monitoring of *L. coffeella* was tested and validated in Brazil, where it is considered the main coffee pest [66]. With the traps, it was possible to monitor adult male populations of *L. coffeella* in the coffee plantation, and by perfecting it with the use of pheromones, it was possible to control high levels of the pest [67]. This method of control is harmless to the environment and avoids the use of chemical insecticides that have caused resistance problems and imbalances in the agro-ecosystem.

**Resistant varieties.** Another leaf miner management strategy is through varietal resistance, which consists of obtaining or selecting coffee varieties that have some secondary metabolite or chemical compound that interferes with the development of the insect (antibiosis) or through plants that produce physical barriers, such as thickness of the leaf or chemical substances that prevent the insect from ovipositing or plant feeding (antixenosis). In the case of the leaf miner, different levels of resistance have been identified in the genus *Coffea* [68]. The species *C. stenophylla* (G. Don), *C. salvatrix* (Swynn. and Philipson), *C. liberica var liberica*, *C. brevipes* (Hiern), *C. jasminoides* 9Welw. ex Hiern),

and *C. farafaganensis* of African origin have been considered resistant, causing high larvae mortality and, as a consequence, reducing the leaf area damage. *Coffea kapakata* (A. Chev.) (Bridson), *C. eugenioides* (S. Moore), *C. racemose* (Lour)., *C. liberica var. dewevrei* (De Wild. and T. Durand) (Lebrun), *C. humilis* (A. Chev.), *C. tetragona* (Jum. and H. Perrier), *C. tsirananae* (J.-F.Leroy), *C. resinosa* (Hook. f.) (Radlk)., *C. millotii* (J.-F.Leroy), *C. bertrandii* (A. Chev.), *C. dolichophylla* (J.-F.Leroy), and *C. bonnieri* (Dubard) are considered moderately resistant, and *C. congensis* (A. Froehner), *C. sessiliflora* (Bridson), *C. travancorensis* (Wight and Arn.) and *C. perrieri* (Drake ex Jum. and H. Perrier) moderately susceptible to *L. coffeella*.

Another possible way to obtain coffee plants with resistance to the leaf miner is through the use of biotechnology via incorporation of genes from organisms other than coffee, such as the cry1Ac gene of the entomopathogenic bacterium *Bacillus thuringiensis*, which is specific for control of Lepidoptera larvae. Perthuis et al. [69] developed genetically modified coffee plants with the addition of a synthetic cry1Ac gene, which exhibited stable resistance to the leaf miner under field conditions for four years in French Guiana.

## 6. The Coffee Red Spider Mite *Oligonychus yothersi* (McGregor) (Acari: Tetranychidae)

*Oligonychus yothersi* (McGregor) is a mite with polyphagous habits. It is distributed in Colombia, Ecuador, Brazil, Argentina, Chile, Costa Rica, the United States (California and Florida), and Mexico [70]. The nymphs and adults scrape and suck the sap content of the leaf cells, drying them out and causing spots in the feeding sites, which later manifest in a characteristic tan coloration on the upper side of the affected leaves [71].

The coffee red spider mite is a seasonal species that occurs in Colombia in prolonged periods of drought and high temperatures. When the rainy seasons arrive, the populations are reduced. Infestations generally begin in coffee lots near highways or uncovered roads, causing dust to settle on the leaves. Likewise, the effect of the deposition of volcanic ash on the coffee foliage has been documented and has a favorable action on *O. yothersi* [72]. The red spider mites produce a silky cover on the upper side of the leaves, and the particles of dust and ash are trapped in the silk cover, offering protection and refuge from predators.

In Colombia, red spider mite populations are naturally regulated by various species of predators: *Stethorus* sp. (Coleoptera: Coccinellidae), *Phytoseiulus* sp. (Acari: Phytoseiidae), and *Chrysoperla* sp. (Neuroptera: Chrysopidae) [73].

### 6.1. Mite Biology

*Oligonychus yothersi* has a high capacity to multiply in a very short time when environmental conditions are favorable. The development time of this mite is directly related to temperature [74]. In this way, at a temperature of 15 °C, 34 days are required for an egg to reach the adult state, while at 20 °C, this occurs in 15.6 days (Figure 17). There is no research that relates environmental humidity to the development of this arthropod. Within the temperature variation, it can be stated that Colombian coffee farming offers the optimal range for the development of the red spider mite (18–22 °C) [74].

### 6.2. Mite Damage

Damage to the leaves is caused by the larvae, nymphs, and adults of the mite when they feed with their mouthparts, which are of the scraping–sucking type, with which they pierce the cells of the epidermis and mesophyll, absorbing the cellular content; consequently, the leaves lose their natural shine and become tanned, with a reduction in leaf area and the photosynthetic capacity of the plant by up to 70% when infestations are high [73] (Figure 18).

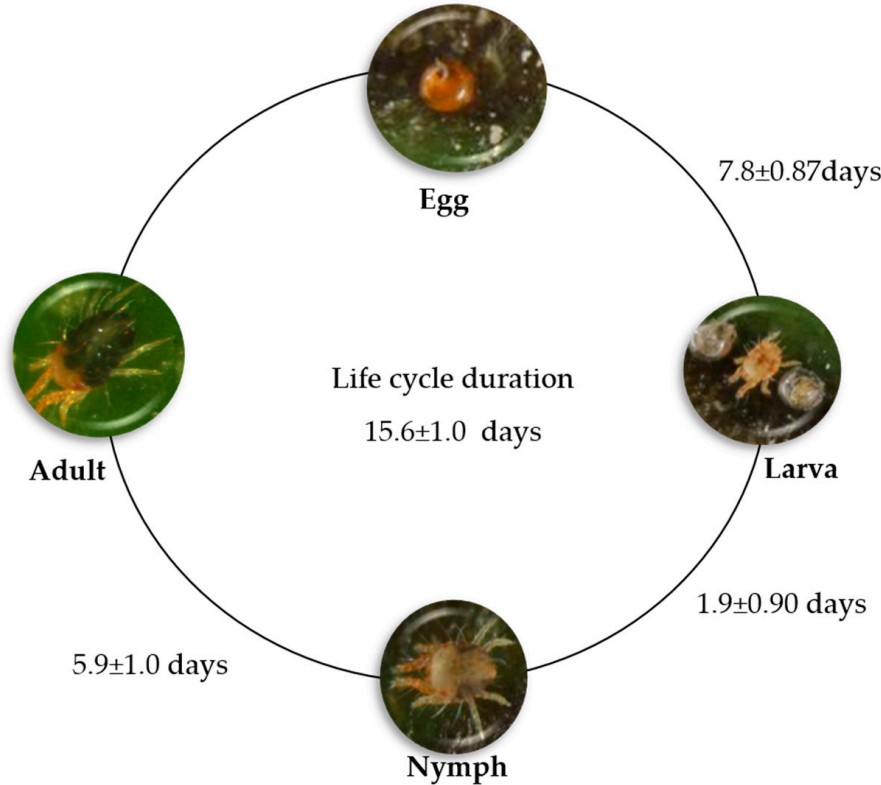

**Figure 17.** *Oligonychus yothersi* life cycle in coffee.

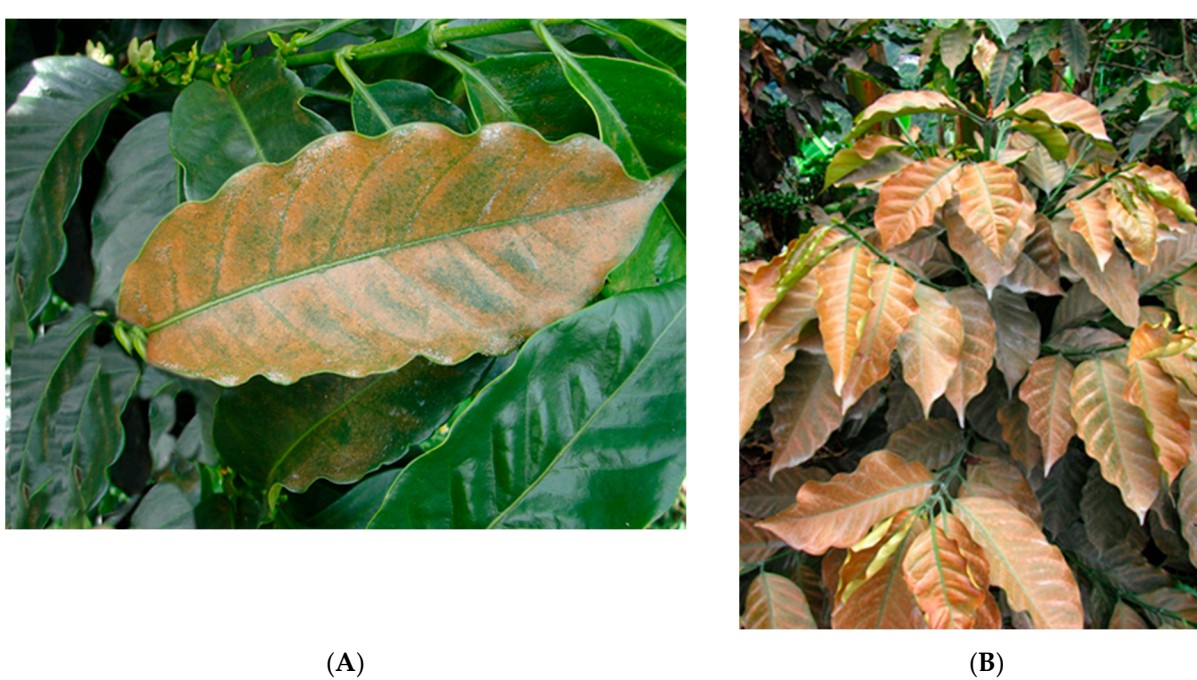

(**A**)                                                                 (**B**)

**Figure 18.** *Oligonychus yothersi* damage. (**A**) Leaf tanned by mite attacks; (**B**) tree totally affected by the red spider mite.

*6.3. Factors Affecting Red Spider Mite Survival*

Climatic factors such as temperature, humidity, rain, drought, and wind influence the behavior, abundance, and distribution of red spider mite populations. Since this mite does not have wings, its mobility depends on the wind, which is the main dispersal factor from one field to another and between farms. Humans can act as a dispersal agent

when moving through the coffee plantations. Temperature is one of the most important environmental factors affecting red spider mite populations [74]. High temperatures promote an increase in the oviposition rate, fecundity, and generational development; the relationship between temperature and development time in days of *O. yothersi* from egg to adult is faster as temperature increases. Therefore, during prolonged periods of drought and high temperatures, populations increase notoriously. The climatic conditions during the rainy seasons are factors of natural mortality of the red spider mite. Since the red spider smite mainly inhabits the upper part of the leaves, which have a smooth texture in coffee, the larvae, nymphs, and adults are easily washed away by the water. In addition, the water deposited on the leaf blade affects the eggs and mobile stages of the mites [73]. This is one of the reasons that explain the low populations of this mite in coffee during wintertime.

### 6.4. Effect of Volcanic Ash Deposition on the Coffee Red Spider Mite

The effect of volcanic ash on the activity of arthropods has been studied in Argentina, Costa Rica, and Colombia after the eruptions of the Puyehue-Cordón, Irazú, Poás, and Nevado del Ruiz volcanoes [72,74,75]. The literature records a favorable effect for certain coffee pests, such as the red spider mite, the mealybug *Planococcus citri* (Risso, 1813) (Hemiptera: Pseudococcidae), and the coffee leaf miner *Leucoptera coffeella* [72,75].Volcanic ash, for other pests, predators, and parasitoids, causes dehydration and death through physical control by acting as an abrasive. In the case of the red spider mite, the habit of producing a silky tissue on the upper side of the leaves allows the particles of dust and ash remain trapped in the silk cover and serve as a defensive barrier and protection against predators.

### 6.5. Red Spider Mite Natural Enemies

Red spider mite populations are naturally regulated by several species of the predators *Stethorus* sp. (Coleoptera: Coccinellidae), the most abundant natural enemy found in association with populations of this mite. This small, black beetle of 1.0 to 1.3 mm in length is observed consuming all biological stages (eggs, larvae, nymphs, and adults) of *Oligonychus yothersi* and is frequently found on a coffee leaf up to five instars between larvae and adults of this predator.

Other species of generalist predators of the red spider mite have been recorded in coffee plantations, such as *Azya orbigera* (Mulsant, 1850), *Cycloneda sanguinea* (Linnaeus, 1763), *Harmonia axyridis* (Pallas, 1773), *Scymnus* sp., *Psyllobora confluens* (Fabricius, 1801), and *Brachiacantha bistripustulata* (Fabricius, 1801) (Coleoptera: Coccinelidae). The predatory mite *Phytoseiulus* sp. (Acari: Phytoseiidae) preys on all states of the red spider mite. This translucent yellow mite shelters in the domatia of coffee leaves, which are cavities along the midrib on the underside of the leaves. Likewise, the larvae and adults of *Chrysoperla* sp. (Neuoroptera: Chrysopidae) exert natural control over the populations of this mite and other predatory insects from the family Staphylinidae (*Oligota* sp.) and Diptera (Syrphidae) [73].

### 6.6. Management Recommendations

1. When the dry weather arrives, it is advised to check the farm to opportunely detect the increase in the populations of red spider mites in the coffee plantation. This increase is manifested by the appearance of trees with leaves of bronze color, especially in the aggregation points.

2. Locate the main spots of red spider mites within the farm.

3. The opportune moment to manage this pest is from the appearance of the first "foci" within the lot. In this way, localized control can be exercised only in the affected areas at a lower cost, and thus, the spread of the pest within the coffee plantation and to neighboring areas can be avoided. It is recommended to not wait for the rains to fall because the population of this mite grows exponentially and causes economic damage in a short time.

    4. Lime sulfur represents an alternative for pest control, especially when acaricides are not allowed, such as in organic crops. It is a home-prepared product made from sulfur and lime and is used to prevent and control some pests, including downy mildew, powdery mildew, botrytis, mites, and thrips. It is recommended to apply it in a concentration of 0.62% for the control of red spider mites. Another option is the use of sulfur 80% WP presented as a wettable powder to be applied as a foliar spray. Both are effective in controlling mites.

    5. When the harvest begins, do not start the coffee harvests from the plots affected by red spider mite since it will carry the plague to the healthy lots.

## 7. Conclusions

    The coffee crop is a perennial crop that have a large impact in the lives of many farmers around the world. It is necessary to equilibrate the economic and social sustainability of the crop with environmental sustainability. Because of that, it is a priority to start evaluating the synergies of different pest control strategies, combining the use of entomopathogens with botanical extracts, push–pull systems, predators, parasitoids, and new volatiles that attract or repel coffee pests and their predators and integrate them into agroecology in such a way that the control of the coffee pests is increasingly efficient and friendly to the environment and society.

**Author Contributions:** Conceptualization, C.E.G., Z.N.G., L.M.C. and P.B. writing—original draft preparation, Z.N.G., L.M.C. and P.B.; writing—review and editing C.E.G. All authors have read and agreed to the published version of the manuscript.

**Funding:** This research received no external funding.

**Acknowledgments:** We would like to acknowledge the librarian Miguel Alfonso Castiblanco for technical support for references organization and review.

**Conflicts of Interest:** The authors declare no conflict of interest.

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
