# Peer review of "Sustainable Strategies for the Control of Pests in Coffee Crops"

_agronomy, doi:10.3390/agronomy13122940_

Round 1
Reviewer 1 Report
Comments and Suggestions for Authors
Agronomy (agronomy-2707981)
Title: Sustainable strategies for the control of pests in coffee crops
This review is an interesting article and the authors intended to bring the updated information on the sustainable control of economically significantly important insect pests of coffee crop. The authors albeit touches variety of control methods against the coffee insect pests but there is a room to further strengthen this review article. This review is timely and an interesting article and will attract good readership. Based on the importance of the article I am of the opinion that this review may be accepted after major revision for its publication in Agronomy journal.
Some of my comments are as follows to be considered by authors for revision:
- I may indicate some grammatically weak sentences and minor spelling mistakes, in my opinion the authors should carefully look and improve the language/spellings for easy understanding of readers
- The authors should not emphasize in abstract on the diseases as this review article is exclusively covering insect pests
- The headings/sub-headings should be consistent for all insect pests in the article, it is better the biology, mode of damage and all non-insecticide control methods like cultural, physical, biological, semiochemicals etc. etc. should be given separately before IPM for all insect pests
- The control options based on the molecular techniques should be included in the article
- The authors should provide ETL and EIL of each insect pest covered in this article
- Its better to provide schematic presentation (original) indicating in detail the biology and if possible the mode of damage of each insect pest in this article
- The authors should include all the relevant literature on the use of entomopathogens against coffee borer especially and other insect pests as well
- The article need an attention of authors as the literature on the combination of entomopathogens/microbes etc. and their interaction with insecticides is published in recent years to control variety of insect pests. I am indicating some examples and I suggest authors should consider them to include in this review to provide a thought for researchers to employ in IPM of coffee insect pests, for example (https://doi.org/10.3390/agronomy12081928; https://doi.org/10.3390/agronomy12051160; https://doi.org/10.1093/jee/toaa209; and many others you may browse)
- The economics of different control methods against each insect pest should be briefly discussed in this review article
- The authors should carefully check the formatting of all the references in the list to ensure the uniformity
Comments on the Quality of English Language
I may indicate some grammatically weak sentences and minor spelling mistakes, in my opinion the authors should carefully look and improve the language/spellings for easy understanding of readers
Author Response
Agronomy-2707981
The authors would like to thank the reviewer 1 for the opinions about the manuscript and all the comments that helped us to improve the quality and clarity of the manuscript. In the new version of the paper:
-All the additions are identified in red color and highlighted in yellow.
-All parts that will be removed from the document appear crossed out and highlighted in yellow.
- all the changes are highlighted in yellow.
Each one of the comments were answered
Reviewer 1
- I may indicate some grammatically weak sentences and minor spelling mistakes, in my opinion the authors should carefully look and improve the language/spellings for easy understanding of readers
Reply: The grammar was review in the whole document.
2.The authors should not emphasize in abstract on the diseases as this review article is exclusively covering insect pests.
Reply: The information about diseases was delete in lines 12,14,18, 49, 50 997.
- The headings/sub-headings should be consistent for all insect pests in the article, it is better the biology, mode of damage and all non-insecticide control methods like cultural, physical, biological, semiochemicals etc. etc. should be given separately before IPM for all insect pests.
Reply: According with the reviewer advice, for each one the insects we leave a short introduction and then, the following subheading:
-Insect biology,
-Insect damage
-Economic damage
-Control strategies
-Then, depending on the insect, others topics were developed.
- The control options based on the molecular techniques should be included in the article.
Reply: The options base on molecular techniques are very interesting however the research in this area is limited and nowadays is not YET used in the coffee fields. It is because of that we will prefer do not included this information and to leave it for future reviews.
- The authors should provide ETL and EIL of each insect pest covered in this article.
Reply: This was included in the Economic damage for each one of the insects.
- Its better to provide schematic presentation (original) indicating in detail the biology and if possible the mode of damage of each insect pest in this article.
Reply: The schematic information was provided for each one of the insects including the damage on the plants. This really help to make the information more understandable. Very good suggestions. There are a new Figure 1 and 2 -line 87- 111, Figure 9 line 397 and Fig 10 line 422, Figure 13 line 580 and figure 14 line 599, Figure 15 line 746, and figure 17 line 894 and Figure 18 line 920).
- The authors should include all the relevant literature on the use of entomopathogens against coffee borer especially and other insect pests as well.
Reply: We add information and the references related to the use of other entomopathogenic fungus such as Metarhizium anisopliae and nematodes (lines 282-299) references 19-20-21-22. However, in this review we want to refer to papers in which a real validation of the strategies has been tested and not only in laboratory assays because the information can be very extensive without validation.
- The article need an attention of authors as the literature on the combination of entomopathogens/microbes etc. and their interaction with insecticides is published in recent years to control variety of insect pests. I am indicating some examples and I suggest authors should consider them to include in this review to provide a thought for researchers to employ in IPM of coffee insect pests, for example (https://doi.org/10.3390/agronomy12081928; https://doi.org/10.3390/agronomy12051160; https://doi.org/10.1093/jee/toaa209; and many others you may browse).
Reply: We have found a large amount of paper that as the reviewer mention related to the use of Insecticides in combination with Entomopathogen. However, in most of the cases the reduction in the use of insecticides is not quite significative. In additions, because the purpose of this review is to show the possibilities of controlling these pests without the use of insecticide at all. We don’t believe that topic correspond to this review.
- The economics of different control methods against each insect pest should be briefly discussed in this review article.
Reply: We include a section in economic damage for each one of the insects and the corresponding references.
- The authors should carefully check the formatting of all the references in the list to ensure the uniformity.
Reply: We did it and we add more references according to the new information that was added.

Reviewer 2 Report
Comments and Suggestions for Authors
First of all, this reviewer congratulates the authors for the work done.
Secondly, some points for improvement in your manuscript are below:
In the first paragraph, it talks about "coffee bags." What unit of weight is it equivalent to? It would be appropriate to indicate it, maybe with a parenthesis for example, in the first appearance on line 37. (A coffee bags = XX Kg or Tons - International System of Units - )
Is there any reference that can validate the information on lines 49-54, 72-73, 194-195, 241-242?
Line 59, you must indicate the abbreviation IPM in parentheses, because it is the first date. They did it on line 74. Ditto with IMW in lines 412 (1st time) and 608.
The feet of the figures should be understandable on their own. If acronyms or abbreviations are used, they should be specified in the figure itself.
Line 77, you must use capital letter in "figure 1".
Line 115, you should write the dot in “pg”.
The images in figure 4 have the legend in Spanish. This reviewer suppose it will be difficult to translate it in the image itself, so It is suggested making a note in the figure footer indicating the translation to facilitate understanding for English speakers or any other formula that is easy for the authors to implement if possible. If they are external images, do not forget please to cite the source too.
At the end of Figure 5 footer, a parenthesis is necessary.
In section 2.1: The title could be simplified since the abbreviation has been indicated above and the authors should use official European names: “European Green Deal” or “Farm to Fork”.
Line 239, the abbreviation IPM could be used. Also in line 294-295.
Line 262, the abbreviation CBB could be used.
Line 656, the abbreviation IMW could be used.
Line 278, the year should be remove before the reference. Also in line 281, 694, 697, 749 and 782.
If there is no confidentiality issue, this reviewer suggests adding a UTM coordinate reference to the location of the Naranjal Cenicafe's Main Experimental Station plot.
Line 415, the last square bracket should be removed.
Figure 8, the "Natural control" square, if it can be enlarged a little, it would be recommended. The names are difficult to read.
It would be interesting if the authors did an entire revision of the document and applied the same criteria for the citations of the living beings that are cited. To understand what this reviewer wants to show, see for example the paragraph between lines 841 and 844, where different criteria are observed when citing them: without parentheses, with parentheses, Linnaeus is cited here when not elsewhere, and so on. To sum up, apply the same criteria.
This reviewer wish you good luck with peer review process and that you can publish your paper soon.
Best regards.
Author Response
Agronomy-2707981
The authors would like to thank the reviewer 2 for the congratulations and all the comments that helped us to improve the quality and clarity of the manuscript. In the new version of the paper:
-All the additions are identified in red color and highlighted in yellow.
-All parts that will be removed from the document appear crossed out and highlighted in yellow.
- all the changes are highlighted in yellow.
Each one of the comments were answered.
Reviewer 2
- In the first paragraph, it talks about "coffee bags." What unit of weight is it equivalent to? It would be appropriate to indicate it, maybe with a parenthesis for example, in the first appearance on line 37. (A coffee bags = XX Kg or Tons - International System of Units - )
Reply: We add the information in line 37.
2.Is there any reference that can validate the information on lines 49-54, 72-73, 194-195, 241-242?
Reply:
Reference for line 49-54 and 72-73 correspond to-
- Bustillo, A.E.; Cardenas, R.; Villalba, D.A.; Benavides Machado, P.; Orozco, J.; Posada, F.J. Manejo integrado de la broca del café : Hypothenemus hampei Ferrari en Colombia; Cenicafé: Chinchiná, Caldas, Colombia, 1998; ISBN 978-958-96554-0-5.
Reference for Line 194-195 – New document line 211-212 correspond to-
- Fetting, C. The European Green Deal; European Sustainable Development Network: Vienna, Austria, 2020; p. 22.
Reference for Line 241-242 -line 227-228 new manuscript, correspond to-
- Benavides Machado, P.; Arévalo, H. Manejo Integrado: Una Estrategia Para El Control de La Broca Del Café En Colombia. Rev. Cenicafé 2002, 53, 39–48.
3.Line 59, you must indicate the abbreviation IPM in parentheses, because it is the first date. They did it on line 74. Ditto with IMW in lines 412 (1st time) and 608.
Reply: In line 60 (IPM) was added. In line 92, 198, 227, 228, 301 only IPM abbreviation was left.
In line 457 (IMW) was added. In line 660, and 708 the abbreviation was only left.
4.The feet of the figures should be understandable on their own. If acronyms or abbreviations are used, they should be specified in the figure itself.
Reply: The feet of all the figure were check again and the changes were done.
- Line 77, you must use capital letter in "figure 1".
Reply: Done line 94.
- Line 115, you should write the dot in “pg”.
Reply: Done pg. 49 Line 132.
- The images in figure 4 have the legend in Spanish. This reviewer suppose it will be difficult to translate it in the image itself, so It is suggested making a note in the figure footer indicating the translation to facilitate understanding for English speakers or any other formula that is easy for the authors to implement if possible. If they are external images, do not forget please to cite the source too.
Reply: Figure 4 in the new manuscript is Figure 6 (Line 158), in this, the Spanish legends were deleted. The explanation of the figure was included in the footer.
- At the end of Figure 5 footer, a parenthesis is necessary.
Reply: Done, it was added in line 189 (Now Figure 7).
- In section 2.1: The title could be simplified since the abbreviation has been indicated above and the authors should use official European names: “European Green Deal” or “Farm to Fork”.
Reply: Done, in Line 206 pack was change to deal.
- Line 239, the abbreviation IPM could be used. Also in line 294-295.
Reply: Done in line 227, 228, 302 Integrated management was change to IPM.
- Line 262, the abbreviation CBB could be used.
Reply: Done, Line 249, 271, coffee berry borer was change to CBB.
- Line 656, the abbreviation IMW could be used.
In line 458 (IMW) was added. In line 660, and 708, the abbreviation was only left.
- Line 278, the year should be remove before the reference. Also in line 281, 694, 697, 749 and 782.
Reply: Done, the years were removed from lines 265, 268, 754, 757, 826, 859.
- If there is no confidentiality issue, this reviewer suggests adding a UTM coordinate reference to the location of the Naranjal Cenicafe's Main Experimental Station plot.
Reply: Information was added lines 305-306 (04°59’ N, 75°39’ W, and an altitude of 1.400 m.a.s.l)
- Line 415, the last square bracket should be removed.
Reply: Done, the bracket was removed from line 460.
Figure 8, the "Natural control" square, if it can be enlarged a little, it would be recommended. The names are difficult to read.
Reply: Figure 8, in the new manuscript Figure 12 (Line540-542) was reorganized and the sentences and title enlarge.
- It would be interesting if the authors did an entire revision of the document and applied the same criteria for the citations of the living beings that are cited. To understand what this reviewer wants to show, see for example the paragraph between lines 841 and 844, where different criteria are observed when citing them: without parentheses, with parentheses, Linnaeus is cited here when not elsewhere, and so on. To sum up, apply the same criteria.
All the insect scientific names and the Linnaesus cited were reviewed and corrected see lines 553-562.

Reviewer 3 Report
Comments and Suggestions for Authors
This review presents a good and updated summary of information about different control methods for some important coffee pest, that would be relevant to setup (or even improve) future integrated pest management strategies in other countries. Particularly, the current knowledge gained on the biology and natural enemies of the different pests, represents an unique information for agronomist and farmers from other producing countries, interested on new control alternatives.
Before publication, some minor modifications seem necessary in order to bring more precision and clearness to the information presented in the document. Please find below some specific suggestions:
1. Introduction:
Lines 56-57: the total number of hectares of coffee and the % of area with CLR resistant varieties used in Colombia should be reviewed. Based on an official presentation of FNC (2022) it is indicated that in Colombia there are 842,000 ha with coffee, and that 86% (724,120 ha) are planted with resistant materials. Please provide the precise information.
Questions: What is the situation of the root nematodes in Colombia? They are not included into the review because it doesn’t represent an important issue for coffee?. Please comment.
Readers would be also interested on some data about the economic damage carried out by each pest. The introduction lacks statistics (i.e. affected areas, incidence levels) about the current economic damage caused by each of the mentioned pests. Please complete.
2. CBB in Colombia:
Figures 1 and 2: the Y axis in the figures lacks the unit. Please add (%) to the scale.
Figure 3. The addition of a reference is recommended.
Figure 4. The scale of vulnerability included in each figure (a, b, c) need to be traduced to English. Please modify.
Line 241: it seems that this is the first time that “Cenicafé” is mentioned in the text. Therefore, please explain its significance = Centro Nacional de Investigaciones del café”.
Figure 6. Even if the strategy is well explained in the text, the figure itself is not clear in terms of the scale and the different periods of control. Please consider the possibility to improve this figure making it clearer and more accessible for readers.
Further, the figure description should also be clear for the readers. Please consider a short explanation taking in account the elements mentioned in the figure.
3. Monalonion velezangeli…
Figure 7 and Figure 8. Same comment than Figure 6. The Figure itself should be clear and understandable for readers without the necessity to go deeply in the text. This figure is not clear for readers. Please consider a modification of the figure and a clearest paragraph explanation.
Author Response
Agronomy-2707981
The authors would like to thank the reviewer 3 for the opinions about the manuscript and all the comments that helped us to improve the quality and clarity of the manuscript. In the new version of the paper:
-All the additions are identified in red color and highlighted in yellow.
-All parts that will be removed from the document appear crossed out and highlighted in yellow.
- all the changes are highlighted in yellow.
Each one of the comments were answering
Reviewer 3
- Introduction:
Lines 56-57: the total number of hectares of coffee and the % of area with CLR resistant varieties used in Colombia should be reviewed. Based on an official presentation of FNC (2022) it is indicated that in Colombia there are 842,000 ha with coffee, and that 86% (724,120 ha) are planted with resistant materials. Please provide the precise information.-
Reply: The changes were done between line 56-59.
Questions: What is the situation of the root nematodes in Colombia? They are not included into the review because it doesn’t represent an important issue for coffee?. Please comment.
Reply: The reviewer is right, nematodes are not a problem in Colombian coffee plantation. In part due to the way that the seedling and nursery system is organized in the country.
Readers would be also interested on some data about the economic damage carried out by each pest. The introduction lacks statistics (i.e. affected areas, incidence levels) about the current economic damage caused by each of the mentioned pests. Please complete
Reply: In order to add the economic damage data and more information about damages, for each insect we leave a short introduction and then, the following subheading:
-Insect biology,
-Insect damage
-Economic damage
-Control strategies
A schematic information was provided for each one of the insects including the damage on the plants. This really help to make the information more understandable. Very good suggestions. There are a new Figure 1 and 2 -line 87- 111, Figure 9 line 397 and Fig 10 line 422, Figure 13 line 580 and figure 14 line 599, Figure 15 line 746, and figure 17 line 894 and Figure 18 line 920).
- CBB in Colombia:
Figures 1 and 2: the Y axis in the figures lacks the unit. Please add (%) to the scale.
Reply: the % was added in the Figure 3 and Figure 4 corresponding to Figure 1 and 2 in the previous manuscript.
Figure 3. The addition of a reference is recommended.
Reply: The information about the reference of Figure 3 (Now Figure 5) is in line 126-128.
Figure 4. The scale of vulnerability included in each figure (a, b, c) need to be traduced to English. Please modify.
Reply: In Figure 4 (Now Figure 6) the Spanish legends were deleted. The explanation of the figure was included in the footer.
Line 241: it seems that this is the first time that “Cenicafé” is mentioned in the text. Therefore, please explain its significance = Centro Nacional de Investigaciones del café”.
Reviewer is right the information was added first time that is mention in line 126.
Figure 6. Even if the strategy is well explained in the text, the figure itself is not clear in terms of the scale and the different periods of control. Please consider the possibility to improve this figure making it clearer and more accessible for readers.
Further, the figure description should also be clear for the readers. Please consider a short explanation taking in account the elements mentioned in the figure.
Reply: Figure 6 (Now Figure 8), between line 301 and 326 is given all the information about this figure.e
- Monalonion velezangeli…
Figure 7 and Figure 8. Same comment than Figure 6. The Figure itself should be clear and understandable for readers without the necessity to go deeply in the text. This figure is not clear for readers. Please consider a modification of the figure and a clearest paragraph explanation.
Reply. Figure 7 and 8 (Now Figure 11 and 12) were modified and add more information to be more clear.
Figure 11. We change the old footer from Explanatory diagram of the behavior of the climate in the regions where there is chamusquina infesting coffee.
To: Explanatory diagram of the behavior of the climate in the regions where there is chamusquina infesting coffee. (1) Increase in the average temperature before the attack ≥ 19 and <22°C. (2). Decrease in relative humidity to less than 55% during the day.(3). Progressive increase in relative humidity up to 95% at night over a period of 30 days. (m= m.a.s.l). Lines 522-526.
Figure 12, the figure was change and re-organized. We change the old footer: Approach to the sustainable management of Monalonion velezangeli in the coffee crop. Interactions between the main components for control.
To: Approach to the sustainable management of Monalonion velezangeli in the coffee crop. Interactions between the main components for control. Lines 541-542.

Round 2
Reviewer 1 Report
Comments and Suggestions for Authors
I have gone through the revised version of review article, the authors significantly improved their article and addressed all the suggestions of review - I am convinced to accept this article for its publication in Agronomy in its current form
Comments on the Quality of English Language
Some minor spell and grammatical corrections shall be made at the type-setting stage